# Targeting vasoactive intestinal peptide-mediated signaling enhances response to immune checkpoint therapy in pancreatic ductal adenocarcinoma

A paucity of effector T cells within tumors renders pancreatic ductal adenocarcinoma (PDAC) resistant to immune checkpoint therapies. While several under-development approaches target immune-suppressive cells in the tumor microenvironment, there is less focus on improving T cell function. Here we show that inhibiting vasoactive intestinal peptide receptor (VIP-R) signaling enhances anti-tumor immunity in murine PDAC models. In silico data mining and immunohistochemistry analysis of primary tumors indicate overexpression of the neuropeptide vasoactive intestinal peptide (VIP) in human PDAC tumors. Elevated VIP levels are also present in PDAC patient plasma and supernatants of cultured PDAC cells. Furthermore, T cells up-regulate VIP receptors after activation, identifying the VIP signaling pathway as a potential target to enhance T cell function. In mouse PDAC models, VIP-R antagonist peptides synergize with anti-PD-1 antibody treatment in improving T cell recruitment into the tumors, activation of tumor-antigen-specific T cells, and inhibition of T cell exhaustion. In contrast to the limited single-agent activity of anti-PD1 antibodies or VIP-R antagonist peptides, combining both therapies eliminate tumors in up to 40% of animals. Furthermore, tumor-free mice resist tumor re-challenge, indicating anti-cancer immunological memory generation. VIP-R signaling thus represents a tumor-protective immune-modulatory pathway that is targetable in PDAC.

Unlike many solid tumor malignancies, pancreatic ductal adenocarcinoma (PDAC) is generally unresponsive to immune checkpoint blockade (ICB) therapies that target molecules such as programmed death-1 (PD-1), programmed death-ligand 1 (PD-L1) or cytotoxic T lymphocyte antigen-4 (CTLA-4)[1,2]. Therapeutic resistance of PDAC to ICB is thought to be partly due to low tumor mutational burden, with the exception of the very small fraction of PDCA patients with tumors that have high microsatellite instability[3]. In addition, the tumor microenvironment (TME) in PDAC is characterized by cancer-associated fibroblasts (CAF) that secrete immunosuppressive proteins and metabolites, abundant regulatory T cells (Tregs), immunosuppressive tumor-associated macrophages (TAMs) and dendritic cells, and limited numbers of functional T cells[4–6]. While several clinical strategies target immunosuppressive cells in several cancers, there is limited improvement in the clinical management of PDAC over the past few years[7]. Recently, results from preclinical and translational clinical studies show that the combination of treatment with agnostic CD40 antibody and gemcitabine/nab-paclitaxel can induce potent

✉e-mail: srra@outlook.com; ewaller@emory.edu

immune-mediated control of PDAC[8–10] but these promising preclinical results are not recapitulated in an early phase clinical trial[11].

VIP is a 28-amino acid long neuropeptide present in the brain, pancreas, colon and lung[12]. Overexpression of VIP and its receptors is previously reported in breast, prostate, and lung cancers, wherein VIP promotes growth and metastasis of tumors[13–17]. Immune cells including T cells have VIP receptors that are upregulated upon T cell activation and respond to VIP receptor signaling by inhibiting activation and proliferation as well as promoting the generation of Tregs and Th2 cells[18–20]. We previously report that inhibition of VIP-R signaling by treating mice with a VIP-R antagonist improves T cell-dependent antitumor response in preclinical models of acute leukemia[21,22], and augments adaptive anti-viral immunity[23,24], observations that lead us to explore whether VIP-R antagonists enhance T cell responses in solid tumor cancer models[25].

Here we hypothesize that paracrine production of VIP production by tumor cells within the PDAC TME functions like an immune checkpoint pathway that limits the antitumor activity of VIP-receptor-expressing T cells and that inhibiting VIP receptor signaling improves T cell-dependent responses to immune checkpoint therapy and survival in preclinical models of PDAC. We investigate these concepts by treating tumor-bearing mice with potent VIP-R antagonists and show that the combination of VIP-R antagonist and anti-PD-1 antibody therapy synergistically enhances activation, decreases exhaustion and recruits tumor-infiltrating T cells in murine PDAC (TIL). Furthermore, combination drug therapy decreases tumor growth rates and results in the complete regression of tumors in a fraction of animals. Reversing VIP-mediated immune suppression of T cells following once-daily administration of a VIP-receptor antagonist suggests broad applications in treating solid tumors and hematological malignancies. Conservation of VIP peptide sequences across species and the activity of VIP-R antagonists in potentiating human T cell activation supports potential clinical translation.

## Results

### VIP is overexpressed in human and murine pancreatic cancer

Expression levels of VIP mRNA across different human tumors were compared using the Cancer Genome Atlas (TCGA). Pancreatic and gastrointestinal cancers had the highest VIP mRNA expression when compared to other solid tumor malignancies (Fig. 1a). Immuno-fluorescence (IF) staining of human PDAC tumors showed increased expression of VIP in pancreatic ductal carcinoma cells with co-expression of cytokeratin-19 (CK19) when compared to adjacent normal tissues (Fig. 1b, Supplementary Fig. 1a). Additionally, analysis of culture supernatants obtained from human and murine PDAC cell lines showed that most PDAC cell lines secrete VIP (Fig. 1c). On the other hand, supernatants from murine melanoma cell lines B16F10 and D4M had low to undetectable VIP (Fig. 1c). The potential for tumor-secreted VIP to have systemic effects on the immune system is supported by the observation that immunocompetent C57BL/6 mice implanted with murine PDAC tumors had significantly elevated levels of plasma VIP when compared to mice with comparable tumor volumes of B16F10 melanoma (Fig. 1d). These findings are in accordance with the human VIP mRNA expression data in which melanoma had low expression of VIP, while PDAC had high levels of VIP mRNA. Similarly, human PDAC patients with pancreatic cancer had significantly higher plasma VIP levels than healthy volunteers (Fig. 1e), suggesting that plasma VIP is a potential biomarker for PDAC. Corroborating this hypothesis, plasma VIP increased linearly with increased volume of KPC-Luc tumors in mice (Fig. 1f).

Intriguingly, orthotopic implantation of KPC-Luc cells resulted in higher plasma levels of VIP than both culture supernatants of KPC-Luc and plasma from mice with subcutaneous KPC-Luc tumors, suggesting that the desmoplastic TME created in the orthotopic model contributes to higher blood VIP levels (Fig. 1d). In support of this,

supernatants from primary CAFs from PDAC patients and that from h-iPSC-PDAC-1, a human pancreatic CAF cell line, secreted high levels of VIP (Fig. 1g). These data confirm expression of high levels of VIP by tumor cells and stromal cells within the TME of human and murine PDAC.

### Inhibiting VIP-R signaling promoted T cell activation while decreasing exhaustion in vitro

Expression of VIP by PDAC and the TME suggested that VIP may function as a paracrine and/or autocrine factor for cancer cell survival. Human PDAC tissues express both VPAC1 and VPAC2[26–28], the two VIP receptors. Additionally, we confirmed VPAC1 and VPAC2 expression in both murine and human PDAC cell lines via western blot (Supplementary Fig. 1b–d). Thus, to test the autocrine effect of VIP made by tumor cells on VIP- receptor expressing tumor cells, we first evaluated whether inhibiting VIP-R signaling affects the growth of PDAC cells in vitro. For these and subsequent studies we used peptide VIP-R antagonists predicted to have greater receptor affinity to human VPAC1 and VPAC2 than VIPhyb based upon in silico modeling. Treatment of PDAC cells with increasing concentrations of ANT008 did not affect viability of PDAC cell lines in vitro, except a transient effect in MT5 (Supplementary Fig. 2a). Notably, growth of KPC-Luc, Panc02, Capan02 and BxPC3 was not affected by addition of VIP-R antagonists (Supplementary Fig. 2a). To test whether VIP produced by PDAC might have an autocrine effect on PDAC growth through VIP-R, we knocked out VPAC2 from the PDAC cell line Panc02 (Supplementary Fig. 2b-d). VPAC2-knockout Panc02 cells had similar growth rates in vitro compared to the wild-type parenteral cell line (Supplementary Fig. 2e). Treatment of VPAC2 knockout or wild-type Panc02 cells with the VIP-R antagonist ANT308 did not inhibit in vitro growth (Supplementary Fig. 2f). VPAC2 KO cells had a slight in vivo growth delay as compared to wild-type cells (Supplementary Fig. 2g) and improved survival (Supplementary Fig. 2h), suggesting possible indirect effects on tumor growth impacted by VIP signaling through the VPAC2 receptor in the Panc02 cell line.

We have previously reported that inhibiting VIP-R signaling with VIPhyb decreases phosphorylation of CREB (phospho-CREB) and enhances T cell proliferation[21]. Here, we evaluated the effect of ANT008 or ANT308 on downstream phospho-CREB signaling leading to higher T cell activation and proliferation. Human T cells activated with anti-CD3 antibody upregulated VPAC1 and VPAC2 expression within 48 h of activation (Fig. 2a, b), with kinetics that are slightly delayed relative to that of PD-1 and CTLA-4, two immune checkpoint molecules targetable with FDA-approved drugs[29] (Fig. 2a, c). To determine the effect of VIP-R antagonists on T cell activation, we measured CD69 expression after treatment with scrambled-VIP-sequence control peptide (Scram), ANT008, or ANT308 using the gating strategy shown in Supplementary Fig. 3. Interestingly inhibiting VIP-R signaling with the addition of VIP-R antagonists significantly increased CD69 expression (Fig. 2d). Notably, ANT308, which has higher predicted binding affinity to VPAC1 and VPAC2 compared to ANT008, significantly increased levels of CD69 in both CD4 and CD8 human T cells (Fig. 2d). Similarly, addition of ANT308 to the cultures more potently inhibited activation-induced phosphorylation of CREB and increased T cell activation[21], when compared to treatment with a control scrambled peptide sequence (Fig. 2e, Supplementary Fig. 4).

We next investigated the effect of VIP-R antagonist on ex vivo expansion of human T cells isolated from PDAC patient peripheral blood. Interestingly, in vitro treatment with ANT008 significantly decreased the proportion of human Tregs (CD4$^+$ CD25$^+$ FoxP3$^+$) assessed after 9 days of T cell expansion (Fig. 2f, g). Additionally, ANT008 also significantly decreased T cells with an "exhausted" phenotype, as measured by the proportion of T cells co-expressing PD1 and Tim-3 or PD1 and Lag-3 or PD1, Tim-3 and Lag-3 in both CD4$^+$ and CD8$^+$ T cell subsets[30–32] (Fig. 2h, i, Supplementary Fig. 5).

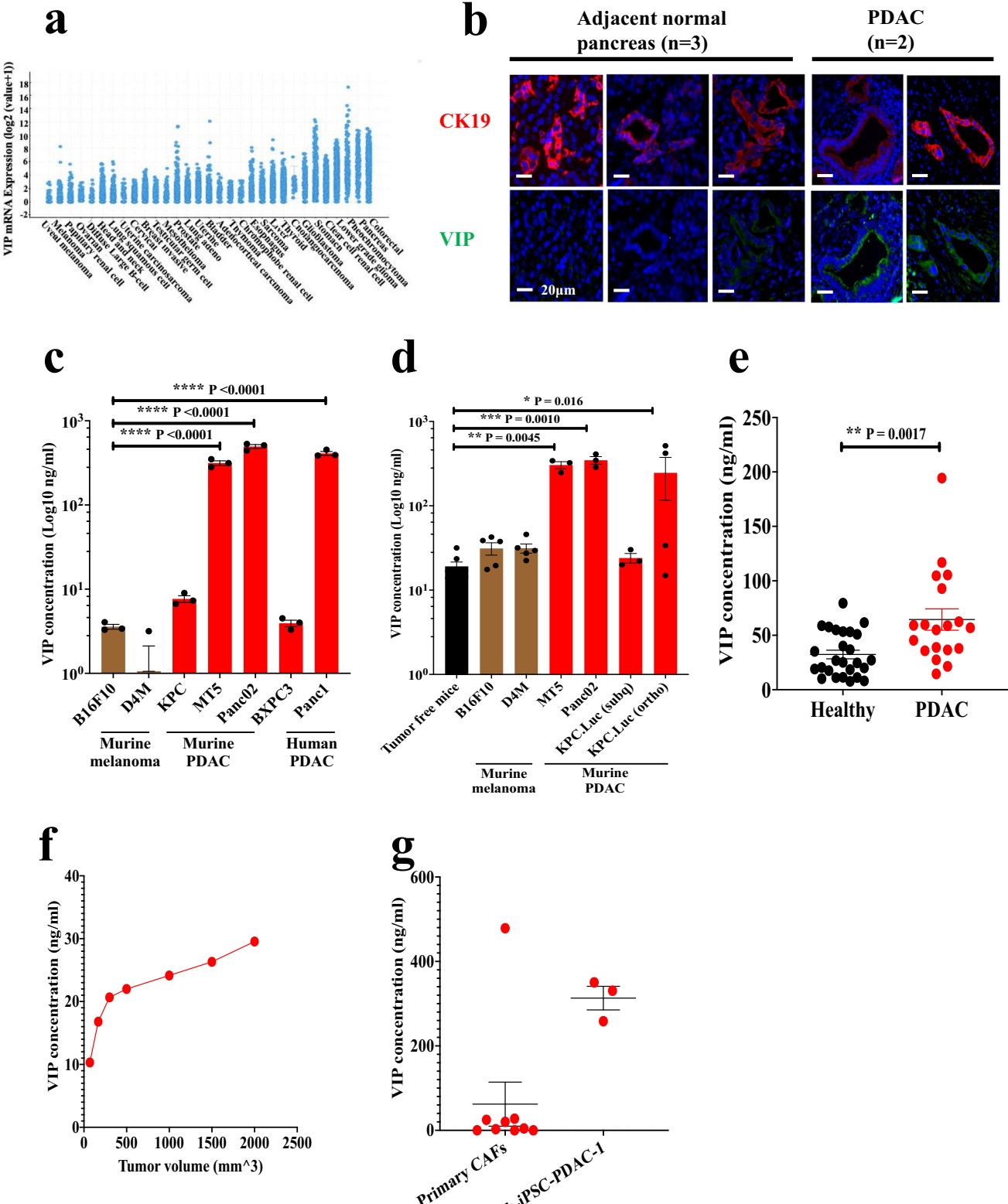

## Antitumor effects of combined blockade of VIP receptor with anti-PD-1 inhibition is T cell-dependent in murine PDAC

To evaluate the effect of VIP-R-mediated signaling inhibition on the growth of PDAC tumors in vivo, we implanted three different murine PDAC tumors (KPC-Luc, MT5 and Panc02) in syngeneic immuno-competent C57BL/6 mice. After tumors were palpable, mice were randomly allocated to treatment with VIP-R antagonists and/or anti-PD-1 monoclonal antibody. VIP-R antagonist treatment in MT5 cells produced a modest decrease in growth in vitro (Supplementary Fig. 6a), and mice bearing MT5 tumors had significantly improved survival and decreased tumor burden following monotherapy with VIP-R antagonist (Fig. 3a and Supplementary Fig. 6a), suggesting direct inhibition of autocrine signaling of VIP through VIP-R on MT5. In all mice bearing either MT5, KPC-Luc, or Panc02 tumors, the combination of VIP-R antagonist with anti-PD-1 significantly decreased tumor burden (Supplementary Fig. 6a–c) and improved

**Fig. 1 | VIP is overexpressed by PDAC. a** VIP mRNA expression levels in various solid malignancies, as obtained from TCGA. **b** Representative images of human PDAC tumors stained with antibodies to VIP (green) and CK19 (red), show higher VIP expression in cancer epithelial cells compared to adjacent normal epithelial cells. Scale bars represent 20μm. Staining of adjacent tissue and PDAC tissue were done in parallel and imaged on the same day. The experiment was performed once. Levels of VIP in (**c**) culture supernatants collected from murine and human PDAC cell lines cultured for 24 h ($n = 3$ per cell line) were compared to culture supernatants from B16F10 and D4M melanoma cells. **d** the plasma of mice bearing

melanoma or PDAC tumors ($n = 5$) compared to plasma of non-tumor-bearing mice; (**e**) plasma of PDAC patients ($n = 19$) compared to that from healthy volunteers ($n = 26$); (**f**) plasma from one C57BL/6 mouse bearing a subcutaneous KPC.Luc tumor bled at different times during tumor growth; and (**g**) culture supernatants from primary CAFs isolated from human PDAC tumors ($n = 9$) and PSCL-12 cell line ($n = 3$). $p$ values shown in panels (**c, d**) were calculated using one-way ANOVA and Dunnett's post-hoc test, where the means were compared to B16F10 or tumor-free mice respectively. $p$ values in e were calculated by a two-tailed student $t$-test. Error bars show mean ± SEM. $^*p < 0.05$, $^{**}p < 0.01$, $^{***}p < 0.001$ and $^{****}p < 0.0001$.

survival when compared to mice treated with scrambled peptide and isotype matched IgG (Fig. 3a, b). Notably in all three tumor models, combination therapy resulted in tumor eradication in a significant proportion of mice (40% in KPC and MT5 tumor-bearing mice and 30% in mice with Panc02 tumors) with no difference in outcome by sex of the mice (Supplementary Fig. 6d, e). In addition to the anti-tumor efficacy, the safety of VIP-R antagonists in immunologically naïve mice was confirmed (Supplementary Fig. 7a–g). Specifically, daily subcutaneous administration of ANT008 or ANT308 at the same dose and frequency as used in the anti-cancer treatment protocol did not affect the overall survival, activity levels or weight of mice (Supplementary Fig. 7a). Analysis of blood samples showed a modest decrease in total leukocytes with treatment (Supplementary Fig. S7b), while analysis of splenocytes did not show any significant differences in frequencies of T, B, NK, dendritic cells (DC), or myeloid-derived suppressor cells (MDSC) (Supplementary Fig. 7b). Furthermore, H&E staining of sections of colon and lung tissue in ANT008- and ANT308-treated mice did not show lymphocytic infiltrates or histopathology to suggest auto-immunity (Supplementary Fig. 7c, d).

We next asked whether the enhanced survival in KPC-Luc tumors treated with VIP-R antagonist and anti-PD-1 is T cell-dependent. The enhanced survival seen with VIP-R antagonist and anti-PD-1 combination therapy was abrogated by depletion of either CD4$^+$ T cells, or CD8$^+$ T cells (Fig. 3c). Combination therapy failed to improve survival in both CD4$^{-/-}$ and CD8$^{-/-}$ tumor-bearing mice (Fig. 3d, e, respectively) indicating the enhanced antitumor response seen with the combination therapy is both CD4$^+$ and CD8$^+$ T cell-dependent.

### Increased T cell activation in tumors of mice treated with a combination of VIP-R antagonist and anti-PD-1

To investigate whether inhibiting VIP-R signaling promotes T cell activation in vivo, we analyzed tumor-infiltrating T cells in subcutaneous KPC-Luc tumors for differences in activation markers. While there were no differences in the proportions of Ki67-, IFN-γ− or IL-4-expressing CD4$^+$ or CD8$^+$ T cells, monotherapy with ANT008 or anti-PD-1 significantly altered the levels of PD-1- and/or Tim-3-expressing T cells. ANT008 monotherapy increased levels of PD-1$^+$ Tim-3$^-$ T cells in CD4$^+$ and CD8$^+$ T cell subsets, suggesting enhanced activation of Tscm (Fig. 4a, b). On the other hand, anti-PD-1 monotherapy increased the frequency of PD-1$^+$ Tim-3$^+$ CD4$^+$ T cells, consistent with T cell exhaustion[30] (Fig. 4a, b). Furthermore, ANT008 or anti-PD-1 monotherapy, as well as combination therapy, significantly decreased the frequency of Tregs within the PDAC tumors (Fig. 4c, d). These findings are in accordance with the effects of VIP-R antagonist on T regs in vitro (Fig. 2f). We further confirmed the effect of ANT008 and anti-PD-1 on T cell activation by analyzing mRNA expression in TIL with Nanostring. While there were no genes significantly upregulated in TIL from mice treated with single-agent ANT008 (Fig. 4e) or anti-PD-1 (Fig. 4f), combination therapy significantly upregulated expression of genes associated with TCR activation and co-stimulation (>4-fold change, FDR < 0.1) (Fig. 4g). Notably, several markers of T cell activation and co-stimulation such as CD27, CD28, CD247, ICOS, TIGIT and CTLA4, were upregulated in the combination group. Cytokines such as IFN-γ, TNF-α and IL2, that

are expressed by activated T cells were also expressed at significantly higher levels in TIL from the combination group (Fig. 4h). Overall, the TCR activation and co-stimulatory pathway score was significantly higher in T cells in the tumors of mice treated with combination therapy when compared to control mice treated with scrambled peptide + isotype IgG (Fig. 4i).

### Combination therapy with VIP-R antagonist and anti-PD-1 induces a tumor-specific T cell response and confers protective immunity to tumor re-challenge

We evaluated if combination therapy increased the frequencies of specific T cell clones and/or antigen-specific T cells within the tumor. DNA was extracted from tumors followed by amplification of TCRß genes. Deep sequencing of the TCRß genes showed increased TCR diversity in tumors from the ANT008 and anti-PD-1 treatment groups ($n = 4$) when compared to control-treated mice (scrambled peptide and isotype IgG; $n = 4$) as shown by Shannon's entropy (Fig. 5a), suggesting more unique TCR responses in the combination group. Upon analyzing the top 50 highest frequency clones in each group, the combination treatment group had the largest numbers of clones shared by at least 2 samples when compared to all other treatment groups (ANT008 and anti-PD-1: 12, scrambled peptide and anti-PD-1: 8, ANT008 and isotype IgG: 6, scrambled peptide and isotype IgG: 6) (Fig. 5b). No significant differences were seen in the frequencies of the shared clones in each treatment group (Fig. 5c).

We next investigated whether combination therapy with VIP-R antagonist and anti-PD-1 promotes tumor-specific T cell responses. MuLV p15E is an antigen expressed on KPC.Luc, Panc02 and MC38 tumors[33] that is considered a tumor-specific antigen due to lack of expression of MuLVp15E in C57BL/6 mice. Antigen-specific CD8$^+$ TILs were enumerated with flow cytometry using a MuLV p15E-H2Kb tetramer reagent. We found that when the tumors were analyzed after 10 days of treatment, tumors from ANT308 and anti-PD-1 treated mice had significantly increased frequencies of tetramer$^+$ CD8$^+$ T cells compared to control treatment tumors (2.85% versus 0.72%, $p < 0.01$; Fig. 5d, e). Together, these data show that combination therapy with VIP-R antagonist promotes tumor-specific T cell responses in KPC.Luc tumors.

The KPC.Luc model has been shown to be partially responsive to single-agent anti-PD1 antibody[34] and we observed some mice without evident KPC.Luc tumors after single agent anti-PD1 treatment (Fig. 3a). To test whether treatment with the combination of VIP-R antagonists with anti-PD1 antibodies enhanced anti-cancer immunological memory more than treatment with anti-PD1 antibodies alone, we rechallenged tumor-free mice (without detectable tumor by palpation or BLI) with a second inoculation of KPC.Luc 80-100 days after initial treatment with either anti-PD1 alone ($n = 6$) or the combination of anti-PD1 antibody with either ANT008 ($n = 5$) or ANT308 ($n = 3$) (Fig. 5f). Mice previously treated with the combination of VIP-R antagonist and anti-PD1 had 100% survival after tumor rechallenge versus 0% long-term survival among mice that were initially treated with single agent anti-PD1 antibody (Fig. 5g). These results substantiate the generation of long-term protective anti-cancer immunological memory following treatment with only the combination of VIP-R antagonist peptides and anti-PD-1 and not with anti-PD1 monotherapy.

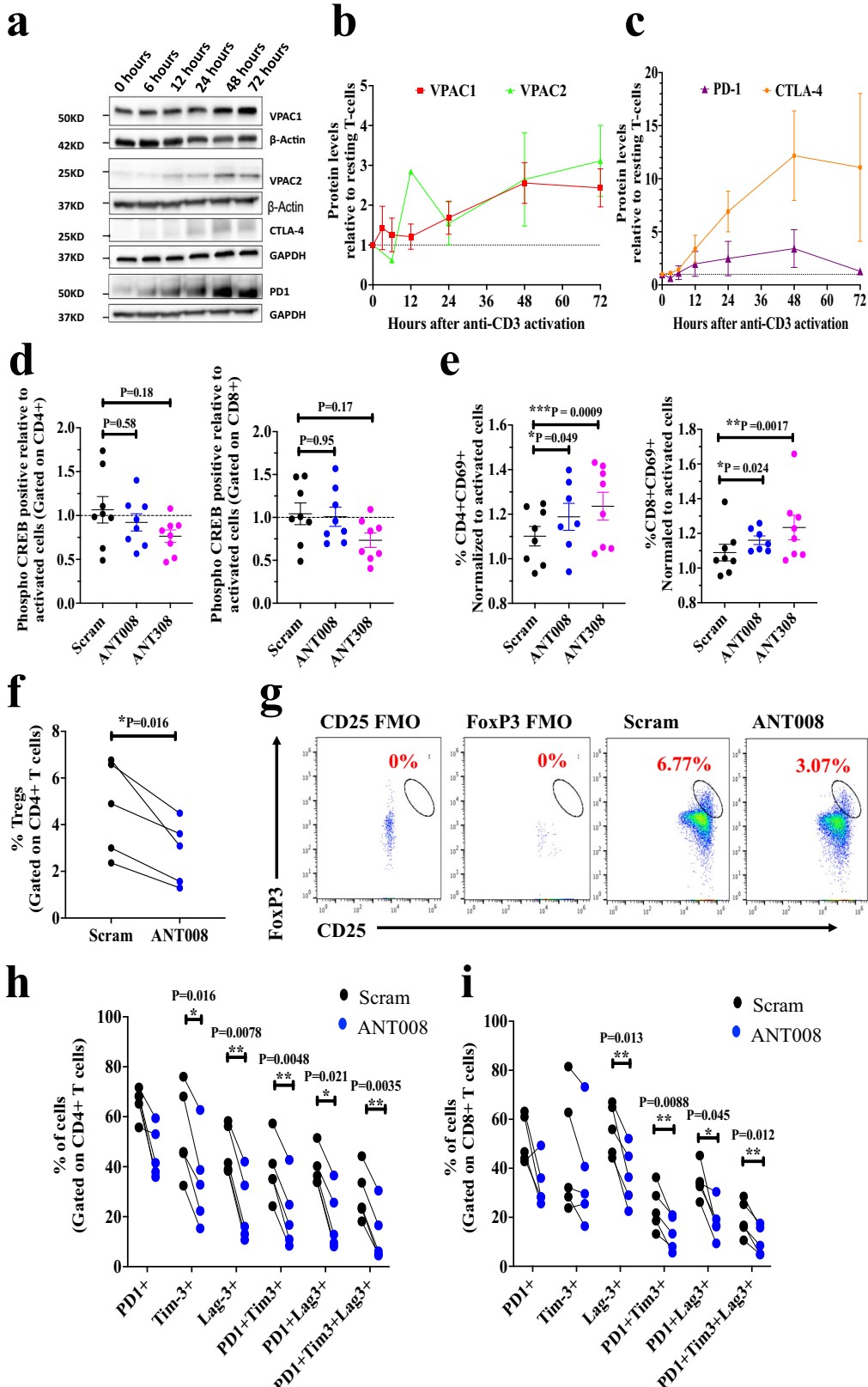

## Synergism between ANT008 and anti-PD-1 decreased tumor burden and increased intra-tumoral T cell frequency in orthotopic murine PDAC

We tested the efficacy of combination ANT008 and anti-PD-1 therapy in a more clinically relevant orthotopic KPC.Luc model, in which PDAC cells are implanted directly into the pancreas, recapitulating some

elements of the TME in clinical PDAC. Bioluminescent imaging (BLI) confirmed successful engraftment of KPC.Luc cells into the tail of the pancreas in wild type mice 6–7 days after implantation. On Day 7 post orthotopic implantation, tumor-bearing mice were randomly assigned to treatment with combinations of scrambled peptide, isotype control antibody, ANT008, or anti-PD-1 MoAb (Fig. 6a). Antitumor responses

**Fig. 2 | Inhibition of VIP-R signaling decreases T cell exhaustion in ex vivo human T cell cultures. a** Representative western blot. The experiment was repeated two times with similar results. **b** quantified expression levels of VPAC1 (*n* = 5) and VPAC2 (*n* = 4 biologically independent samples); and (**c**) PD-1 (*n* = 2 biologically independent samples) and CTLA-4 (*n* = 5 biologically independent samples) in lysates of healthy human T cells expanded with plate-bound human anti-CD3 antibodies for 0, 3, 6, 12, 24, 48 and 72 h. Percentage of (**d**) phosphorylation of CREB (phospho-CREB) downstream of VPAC1/2 receptor at 6 h (*n* = 8 biologically independent samples) and (**e**) CD69 expression at 24 h post activation in CD4+ and CD8+ T cells normalized to levels in control with no peptide (*n* = 8 biologically independent

samples). PDAC patient peripheral blood T cells (*n* = 5 biologically independent samples) were expanded for 9 days with plate-bound human anti-CD3 antibodies ± ANT008 and the (**f**) percentage of Tregs was quantified using the (**g**) gating strategy shown. Percentage of PD1+, Tim-3+, Lag3+ PD1+, Tim-3+, PD1+ Lag3+ and PD1+ Tim-3+Lag-3+(triple positive) in (**h**) CD4+ and (**i**) CD8+ subsets is shown. Statistical differences shown in panels (**d**, **e**) were calculated via mixed effects ANOVA followed by Dunnett's post-test where each sample in the treatment group was compared to the matched sample in the control group (scrambled peptide treated). Statistical differences in panels (**f**), (**h**), (**i**) were calculated via two-tailed paired student *T*-test. Error bars show mean ± SEM. *$p < 0.05$, **$p < 0.01$ and ***$p < 0.001$.

were the greatest in the combination ANT008 plus anti-PD-1 group, such that tumors regressed in 7–11 mice (63%) versus 5/9 mice in the group receiving anti-PD-1 monotherapy (55%) and 4/10 mice in the ANT008 monotherapy group (40%) (Fig. 6b). In addition, combination of ANT008 and anti-PD-1 had a synergistic effect (as shown in Supplementary Table 3) and led to slower growth of tumors and the lowest tumor burden (measured by the weight of the pancreas on day 25) when compared with control mice (Fig. 6c, d). Also, one out of 10 mice each in the anti-PD-1 monotherapy and ANT008/anti-PD-1 combination treatment groups were tumor-free as judged by histological analysis of serial H&E-stained tissue sections. The accuracy of the bioluminescent signal (tumor flux) from the tumors was validated by comparing IVIS and MRI images with cross sectional images of H&E-stained paraffin embedded pancreatic tissue, obtained at the time of necropsy (Supplementary Fig. 8a) and by correlating tumor BLI flux with weight of the pancreas (Supplementary Fig. 8b).

Finally, immunohistochemical analysis of pancreas at Day 25 of therapy evaluated collagen and infiltrating T cells across tumors from mice in the different treatment groups (Fig. 6e). Bands of collagen were visualized in all tumors, consistent with evidence of desmoplastic TME characteristic of human PDAC (Fig. 6e, Supplementary Fig. 8c). Interestingly, tumors from mice that received the combination therapy had significantly higher intra-tumoral levels of CD4+ and CD8+ T cells (Fig. 6f, g), as well as a higher proportion of Ki67+ CD4+ and CD8+ T cells (Fig. 6h, i). Furthermore, there was a positive correlation between T cell density and pancreas weight, such that the smallest tumors in the combination group had higher numbers of proliferating T cells (Supplementary Fig. 8d–g). These findings suggested that combination therapy with ANT008 and anti-PD-1 not only leads to enhanced T cell activation, but also promotes T cell infiltration into the collagen-rich TME of orthotopic murine PDAC tumors.

**Combination therapy with VIP-R antagonist and anti-PD-1 promotes T cell homing into tumors and decreases CXCR4 expression on T cells in tumor-draining lymph nodes**

To test the hypothesis that the enhanced anti-tumor response with combination therapy could be due to increased infiltration of T cells into the TME, we adoptively transferred immunologically naïve GFP+ T cells from C57Bl/6 EGFP-transgenic mice to wild-type C57BL/6 mice bearing KPC.Luc tumors implanted subcutaneously and measured infiltration of GFP+ T cells into the tumor (Fig. 7a). Intriguingly, following three days of treatment with ANT308 and/or anti-PD1 MoAb, mice treated with combination therapy had increased numbers of GFP+ T cells within the tumor when compared to all other treatment groups (Supplementary Fig. 9a–c). Fluorescent microscopy of DAPI stained frozen tumor sections further confirmed significantly increased GFP+ T cell infiltration with combination therapy, that infiltrated the entire tumor (Fig. 7b).

Preclinical and clinical studies have previously shown that down-regulation of CXCR4 promotes mobilization and increases intra-tumoral T cell infiltration in human and murine PDAC tumors[35,36]. We thus asked whether combination therapy with VIP-R antagonist and anti-PD1 modulates expression levels of CXCR4 on T cells. Interestingly, while anti-PD-1 monotherapy increased proportions of Ki67 or

CD69 expressing CD4+ and CD8+ T cells, a significant proportion of activated or proliferating cells also expressed CXCR4 (Fig. 7c, d). On the other hand, treatment with the combination of anti-PD-1 plus VIP-R antagonist increased the proportion of activated CD69+ CD4+ or CD8+ cells with decreased CXCR4 expression (Fig. 7c, d). As AMD3100, a CXCR4 antagonist, has been clinically evaluated as a strategy to mobilize CD8+ T cells into PDAC tumors[37], we next tested treatment with AMD3100 combined with anti-PD-1 and/or VIP-R antagonists. Combination therapy with VIP-R antagonist and anti-PD-1 was superior to the combination of AMD3100 and anti-PD-1 (Fig. 7e), resulting in complete regression of tumors in 20% and 10% of the mice, respectively (p = NS) (Fig. 7f). Interestingly, when all three drugs (VIP-R antagonist, anti-PD-1, and AMD3100) were used in combination, survival was not significantly better than in control mice receiving scrambled peptide, isotype matched IgG and PBS (Fig. 7e).

## Discussion

The clinical efficacy of immune checkpoint blockade in pancreatic cancer targeting PD-1 and CTLA-4 has been modest, despite the remarkable success of ICB in treatment of patients with other solid tumor malignancies[4,38,39]. Clinical and preclinical studies have shown that the poor responsiveness of PDAC to ICB is largely due to an immunologically cold TME characterized by limited numbers of T cells in the tumor parenchyma, and multiple mechanisms that restrict intra-tumoral T cell activation[40–42]. Therefore, strategies that 'boost' T cell priming, or activation may promote enhanced T cell-mediated anti-tumor responses and improve responsiveness to anti-PD-1 or anti-CTLA-4 ICB[8,9]. Using preclinical models of murine PDAC we tested whether treatment with VIP-R antagonists promote activation and proliferation of antitumor T cells and whether these drugs synergize with anti-PD-1.

Our results with selective depletion of CD4+ or CD8+ T cells (Fig. 3c–e) and the generation of cancer-antigen specific immunological memory (Fig. 4e, g) suggest that the dominant effect of VIP-R antagonists is via inhibition of paracrine signaling of VIP produced by tumor cells on T cells that express the VIP-R. While knockout of VPAC2 in Panc02 cells suggested indirect effects of VIP-signaling in vivo, mice with VPAC2 knockout Panc02 tumors had a median survival benefit of 7 days compared to wild type tumors (Supplementary Fig. 2h). In contrast, the combination of VIP-R antagonist and anti-PD-1 synergistically improved T cell-dependent antitumor responses in mice with PDAC, resulting in tumor elimination in up to 40% of treated tumor-bearing mice. Furthermore, since rechallenging the tumor free mice resulted in complete tumor rejection in mice that received the combination of VIP-R antagonist and anti-PD-1, it further emphasized the role of the enhanced adaptive and long-lasting anti-tumor immunity generated in response to inhibiting VIP-R signaling. Recipients of VIP-R antagonist peptide/anti-PD-1 combination therapy had increased homing, activation and proliferation of intra-tumoral CD4+ and CD8+ T cells and marked increases in tumor-antigen-specific T cells within the TME. These findings are in accordance with studies by Vonderheide et al., wherein induction of a robust T-cell response via CD40-signaling overcomes refractoriness of PDAC tumors to ICB[8].

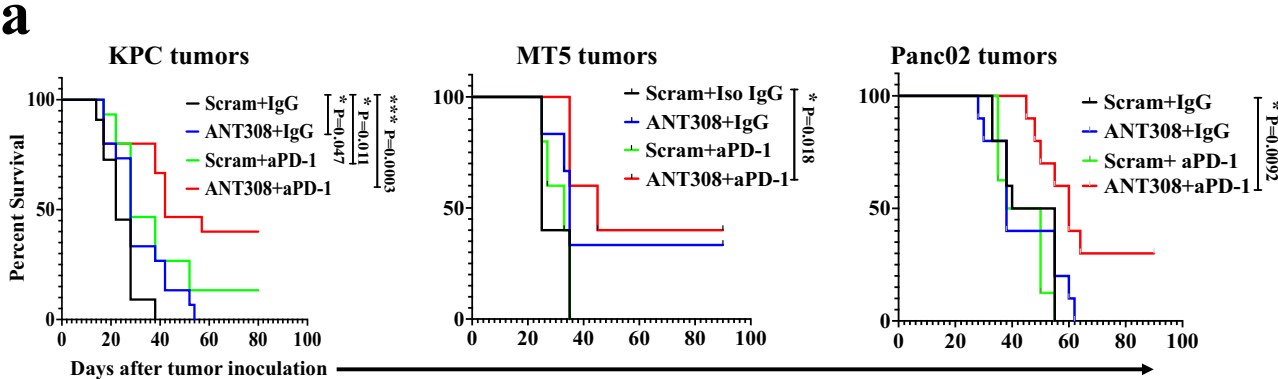

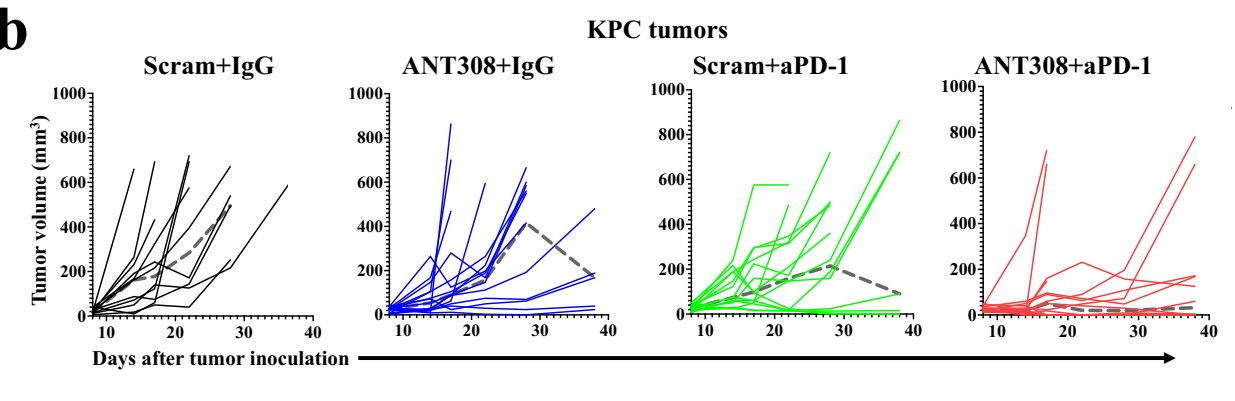

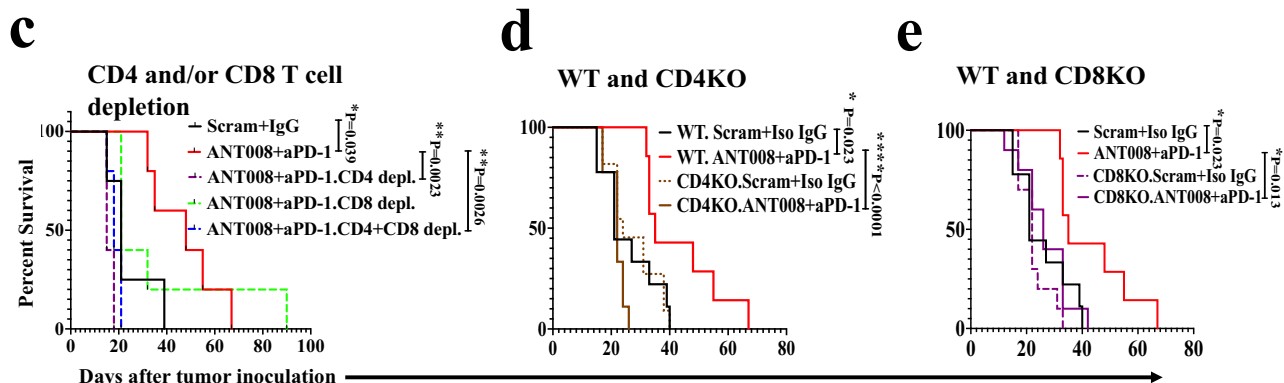

**Fig. 3 | Improved survival in PDAC-bearing mice treated with the combination of VIP-R antagonists and anti-PD-1 is T cell-dependent. a** Kaplan–Meier survival plots of C57BL/6 mice with subcutaneously implanted KPC.Luc, MT5 or Panc02 tumors are stratified by treatment. Both male and females were used in equal proportions for experiments utilizing subcutaneous implantation of KPC-Luc cancer cells. Female mice were used for experiments involving subcutaneosu implantation of MT5 and Panc02 cancer cell lines. **b** Spider plots for KPC.Luc corresponds to results in panel (**a**) as measured by Vernier calipers following subcutaneous tumor implantation in 4 different treatment groups. Median tumor volumes are represented as a dashed gray line (---). In panels (**a**, **b**), tumor cells were implanted in female or male mice with females receiving 10 ug ANT308 and males receiving 20 μg of ANT308 due to higher male mice body weight compared to female mice. Kaplan–Meier survival plots of (**c**) C57BL/6 mice receiving monoclonal CD4 and/or CD8 monoclonal antibodies (**d**) CD4KO or (**e**) CD8KO mice compared to wild-type CD57BL/6 mice with subcutaneously implanted KPC.Luc tumors, stratified by treatment. Statistical differences for Kaplan–Meier curves are calculated via Log-rank test. *$p < 0.05$, **$p < 0.01$ and ***$p < 0.001$, ****$p < 0.0001$.

Exclusion of T cells from the TME is an important characteristic of PDAC tumors that is considered a consequence of a dense desmoplastic stroma[43] containing robust immunosuppressive cells and soluble factors that limit T cell activation[44]. In multiple studies components of the stroma such as cancer associated fibroblasts (and the cytokines and chemokines that they secrete), suppressing effector functions of T cells results in an immunologically "cold" tumor[44–46]. The role of the CXCR4/CXCL12 axis in T cell infiltration in PDAC tumors is complex[35]. CXCR4 expression facilitates homing of I T cells to lymph nodes with high CXCL12 levels where priming to tumor antigens may occur[47]. However, CXCR4+ T cells can be "trapped" in the peri-tumoral extracellular matrix by binding to CXCL12 expressed by CAF and tethered onto KRT19[37]. Thus, high expression of CXCR4 is a predictive marker for poor survival in PDAC patients, and treatment with CXCR4 antagonists increase CD8+ T cell infiltration in the TME[35,36,48]. Consistent with those data, the current findings of synergy between VIP-R antagonists and anti-PD1 align with modulation of CXCR4 levels. Treatment with anti-PD-1 increases expression of CXCR4 on intratumoral T cells[49] while the addition of VIP-R antagonist to anti-PD-1 significantly decreased CXCR4 expression on T cells potentially

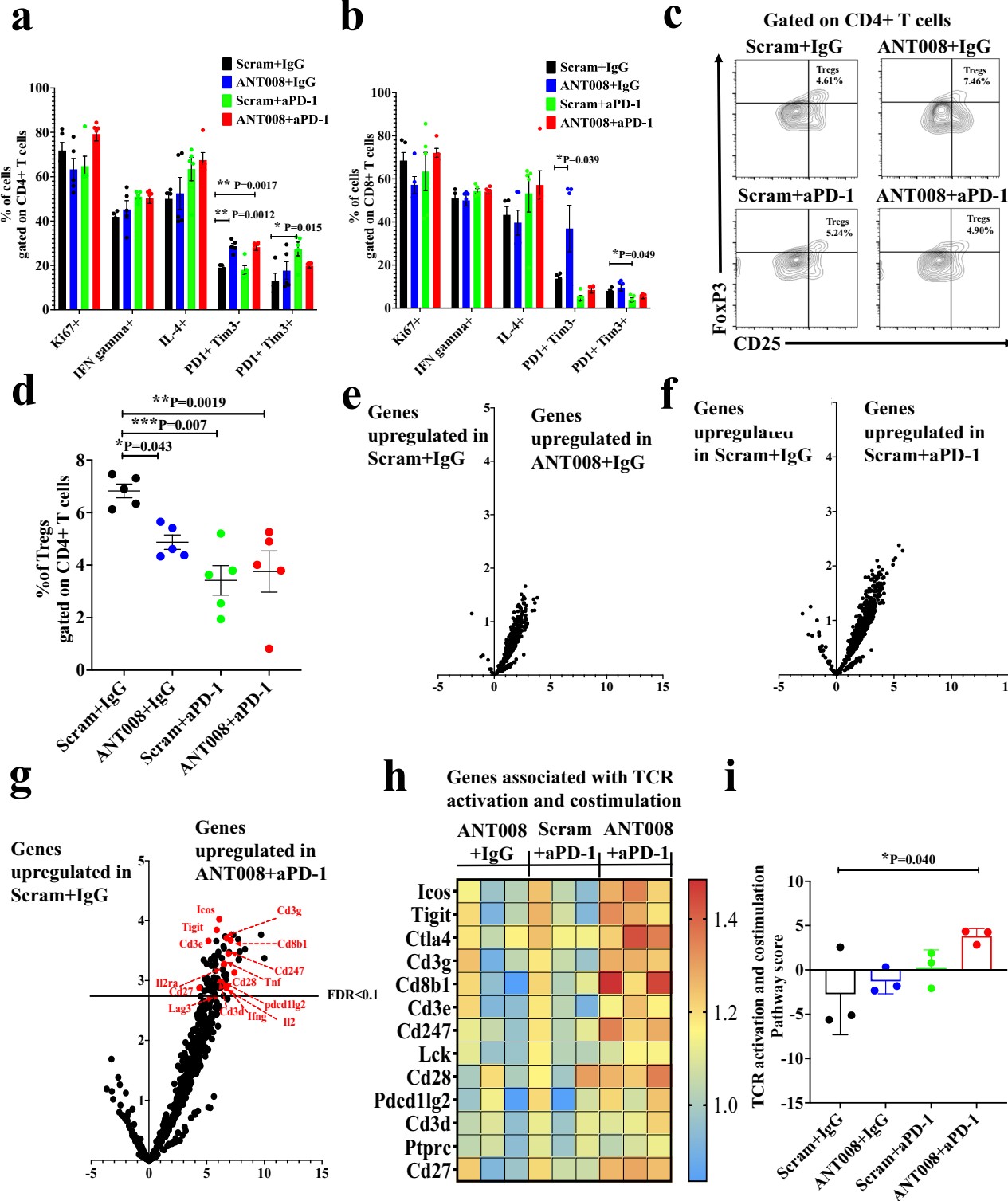

Fig. 4 | Increased T cell activation and reduced frequency of Tregs in KPC.Luc tumors treated with a combination of VIP-R antagonist and anti-PD-1. Sub-cutaneous KPC.Luc tumors in female C57BL/6 mice treated with ANT008 and/or anti-PD-1 ($n = 5$ per treatment group), were analyzed via flow cytometry 10 days after treatment for proportion of (a) CD4+ and (b) CD8+ T cells expressing Ki67, IFN gamma, IL-4, PD-1 and Tim-3. c Representative flow plots showing the gating strategy used to quantify CD25+ FoxP3+ Tregs. d Percentage of Tregs in tumors of the different treatment groups ($n = 5$ per treatment group). Volcano plot showing differential expression of genes in T cells from (e) ANT008 + isotype IgG (IgG) vs scrambled peptide (Scram) + isotype IgG, (f) scrambled peptide + anti-PD-1 vs scrambled peptide + isotype IgG and (g) ANT008 + anti-PD-1 vs scrambled peptide + isotype IgG ($n = 3$ mice per treatment group). Horizontal black line represents false discovery rate (FDR) < 0.1. Genes that are associated with TCR activation and co-stimulation and are at levels significantly higher when compared to Scram+ isotype IgG (FDR < 0.1) are labeled in red. h Heat map showing gene expression changes in genes associated with TCR activation and co-stimulation. i TCR activation and co-stimulation pathway score between the T cells in tumors of mice from the different treatment groups. Statistical differences shown in panels (a, b, d, i) were calculated via ANOVA followed by Dunnett's post-test. Error bars show mean ± SEM *$p < 0.05$, **$p < 0.001$, ***$p < 0.0001$.

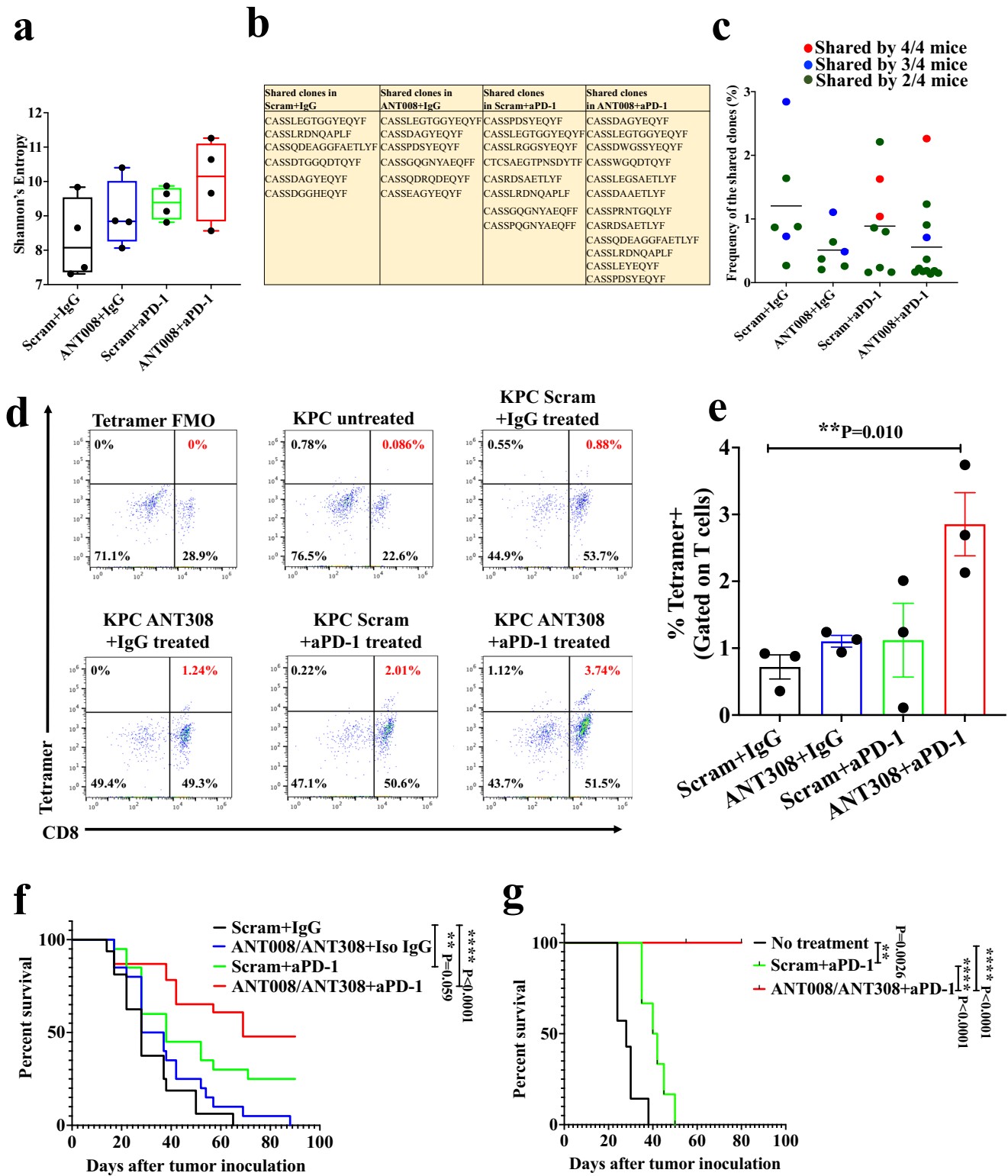

preventing T cells from being "trapped" in the extracellular matrix. Surprisingly, treatment with a triple combination of VIP-R antagonist, anti-PD-1 and a CXCR4 antagonist resulted in similar tumor growth rates and survival to control mice receiving no drugs (Fig. 7e). These data suggest that downregulation of CXCR4, but not complete blockade, could be a superior therapeutic strategy when using a VIP-R antagonist with anti-PD1 to promote T cell trafficking and cytotoxicity within the TME. The apparent antagonism seen when treatment with a VIP-R antagonist was added to the combination of anti-PD1 and CXCR4

antagonists suggests that impactful clinical translation combining these three classes of drugs together would be challenging. In support of our over-arching hypothesis, TIL in mice receiving combination drug therapy expressed significantly higher levels of transcripts associated with TCR signaling and activation, and increased mRNA levels of Th1 cytokines TNF-α, IFN-γ, and IL-2. Furthermore, upregulation of mRNA levels for chemokines CCL2, CCL4 and CCL5 in T cells and the chemokine receptor CXCR4 suggest that intra-tumoral T cells activated by combination VIP antagonist/anti-PD1 therapy may promote

**Fig. 5 | Combination therapy with VIP-R antagonist and anti-PD-1 increased frequency of tetramer+, and CD8+ T cells within the tumor and provide protective immunity to tumor re-challenge. a** Box and whiskers plot showing minimum to maximum Shannon's Entropy in T cells of KPC.Luc tumors in each treatment group. The mid-line represents the median value for each group. **b** List showing TCR-ß amino acid sequences shared between samples of each treatment group and (**c**) the frequencies of the shared clones in each treatment group. Sequences are color-coded to represent number of mice per group (n = 4) that share the specific TCR-ß clone. CD8+ in subcutaneous KPC.Luc tumors were stained with MuLV p15E-H2Kb tetramer after 10 days of treatment with ANT308 and/or anti-PD-1 (n = 3 per treatment group) using the (**d**) gating strategy and (**e**) quantified. **f** Kaplan–Meier survival curves of subcutaneous KPC.Luc bearing mice treated with ANT008/ANT308 and/or anti-PD-1 from day 3-12 after tumor implantation (n = 16 per scrambled peptide + isotype IgG, n = 20 in ANT008/ANT308 + isotype IgG and in scrambled peptide + anti-PD-1 treatment groups; n = 23 in ANT008/ANT308 + anti-PD-1 treatment group). **g** Kaplan–Meier survival curves of tumor-free mice from panel f that were re-challenged with KPC.Luc tumors on the opposite flank (n = 6 in scrambled peptide + anti-PD-1 treatment group; n = 8 in ANT008/ANT308 + anti-PD-1 treatment group). Immunologically-naïve C57BL/6 mice (with no prior tumor exposure or treatment) were inoculated with tumor cells at the same time of the rechallenge (n = 7). Statistical differences shown in panels (**a**, **e**) were calculated via ANOVA followed by Dunnett's post-test, and in panel **f**, **g** were calculated using Log-rank test. Error bars show mean ± SEM *p < 0.05, **p < 0.01, ****p < 0.0001.

recruitment of additional T cells from blood into the TME. This hypothesis is supported by increased accumulation of immunologically naive GFP+ T cells into tumors of mice treated with combination therapy.

Our data suggest VIP-R antagonists act by blocking an inhibitory signaling pathway that limits the activation, proliferation, and survival of immune cells[50,51]. Of note, earlier studies demonstrating increased interferon-gamma production by NK cells following treatment of mCMV-infected mice with a VIP-R antagonist[23] suggests that NK cells may also contribute to the anti-cancer immune responses seen in tumor-bearing mice treated with a VIP-R antagonist. Reduced graft-versus-leukemia effects seen in leukemic mice transplanted with allogeneic marrow and splenocytes from beige donors further supports the contribution of NK cells to the anti-cancer immune effects observed following treatment with VIP-R antagonists[52]. Decreased CREB phosphorylation[22] following in vitro treatment of human T cells with VIP-R antagonists suggests that enhanced NF-κB signaling is responsible for the enhanced T cell activation seen in the murine PDAC models[53]. One of the safety concerns in the use of any ICB is induction of autoimmunity[54]. Notably, treatment of wild type mice with 10 days of daily subcutaneous injections of the ANT008 or ANT308 VIP-R antagonists was not associated with histopathological evidence of autoimmunity, consistent with the absence of auto-immune disease in VIP knockout mice[55] and no evident effect on behavior[56,57]. Thus, the salubrious effect of VIP-R antagonist treatment on anti-cancer immunity in the PDAC models is likely due to local effects of VIP in the TME where VIP is pathologically overexpressed.

This study has some limitations, primarily that murine models used to-date employed transplantable PDAC cell lines that do not fully recapitulate the stromal desmoplasia of clinical PDAC[58]. Nevertheless, mouse T cells penetrated deep into orthotopically implanted tumors treated with the combination of VIP-R antagonists and anti-PD1, apparently crossing dense stromal bands of collagen. Secondly, we limited our focus to a single tumor histology, in a single animal species. Additional models including large animal species (i.e., dogs[59]) should be evaluated. Thirdly, while the current study focused on the treatment of PDAC, several other cancers may also be potential targets for VIP-R antagonists. Published studies indicate antitumor activity of VIP-R antagonists in murine models of myeloid leukemia and lymphoma[21,22], and other cancers over-express VIP[60]. Fourth, the focus and conclusions of the current study are orthogonal to previous studies that examined the suppression of autoimmunity by native VIP, or studies that examined the activity of VIP-R antagonists as tumor cytostatic drugs[14,15,61,62].

Finally, why hasn't VIP signaling been previously identified as a targetable ICB pathway? Pharmacologic antagonism of VIP receptors was explored as a strategy to block autocrine signaling of VIP expressed by tumor cells that stimulates tumor growth[61–65]. The human pancreatic cancer cell line CAPAN-2 expresses VIP receptors and growth is stimulated by VIP[66,67]. VIP antagonists inhibit c-fos mRNA induction by VIP and retard the growth of CAPAN-2 cells in nude mice indicating that VIP receptor antagonists have a tumor-

intrinsic cytostatic effect[63,66]. Preclinical studies explored VIP-R antagonists as cytostatic anti-cancer drugs[16,61–63] but did not lead to clinical trials of VIP-R antagonists in humans, or to studies testing the potential of VIP-R antagonists to augment adaptive immunity. While VIP was identified more than four decades ago[68], the effects of VIP on immunity were studied by Delgado[69,70], and Smally[71] starting in 1999 and leading to a 2006 publication that defined the ability of VIP to induce tolerogenic dendritic cells and Tregs that limit autoimmunity and GvHD[72]. We used VIP-R antagonists designed to have increased affinity to VIP-R which may have contributed to the enhanced immune cell activation reported herein. While the focus of the current study has been on the effect of anti-PD1 antibody and VIP-R antagonists on anti-tumor T cells, this therapy also has the potential to affect immune-suppressive myeloid cells in the TME and block the generation of tolerogenic dendritic cells[73] and affect antigen presentation by tumor-infiltrating macrophages[74] and dendritic cells in tertiary lymphoid structures within the tumor[6,75].

Based upon findings from the current study, we suggest that VIP-R antagonists may represent a tractable approach in the treatment of PDAC[76]. Of note, VIP amino acid sequence is identical between humans and mice, and VIP-R sequences are highly conserved[77], suggesting the VIP-R antagonists with immunological activity in tumor-bearing mice may have comparable properties in human patients. In support of this notion, ex vivo treatment of human T cells isolated from the blood of PDAC patients with VIP-R antagonists promoted T cell activation, downregulation of PD-1, Tim-3 and Lag-3 immune checkpoint molecules associated with immunological senescence and decreased the frequencies of regulatory T cells (Fig. 2f). The ability of small peptide-based drugs to diffuse into the stromal-rich TME of PDAC and promote the in situ activation of cancer-specific T cells may be an advantage of targeting the VIP::VIP-receptor axis in immuno-oncology. Furthermore, overexpression of VIP may represent a biomarker useful to identify patients with PDAC and other cancers sensitive to the ICB activity of VIP-R antagonists (Fig. 1b, e), analogous to the use of PD-L1 staining of cancer as a predictive biomarker for response to anti-PD1 ICB. Further clinical development of this concept will require appropriate preclinical pharmacokinetic and toxicology studies.

## Methods

The research complies with all relevant ethical regulations. Studies involving human samples were approved by the Emory Institutional Review Board (IRB). All experimental procedures involving animals were approved by the Institutional Animal Care and Use Committee (IACUC) at Emory University.

### Cell lines and reagents

MT5 and KPC.Luc cells were generous gifts from Dr. Tuveson (Cold Spring Harbor Laboratory, Cold Spring Harbor, NY) and Dr. Logsdon (MD Anderson Cancer Center, Houston, Texas), respectively and Panc02 cells were provided by Dr. Pilon-Thomas (H. Lee Moffitt Cancer Center, Tampa, FL). BXPC3, Panc1 and B16F10 were from ATCC

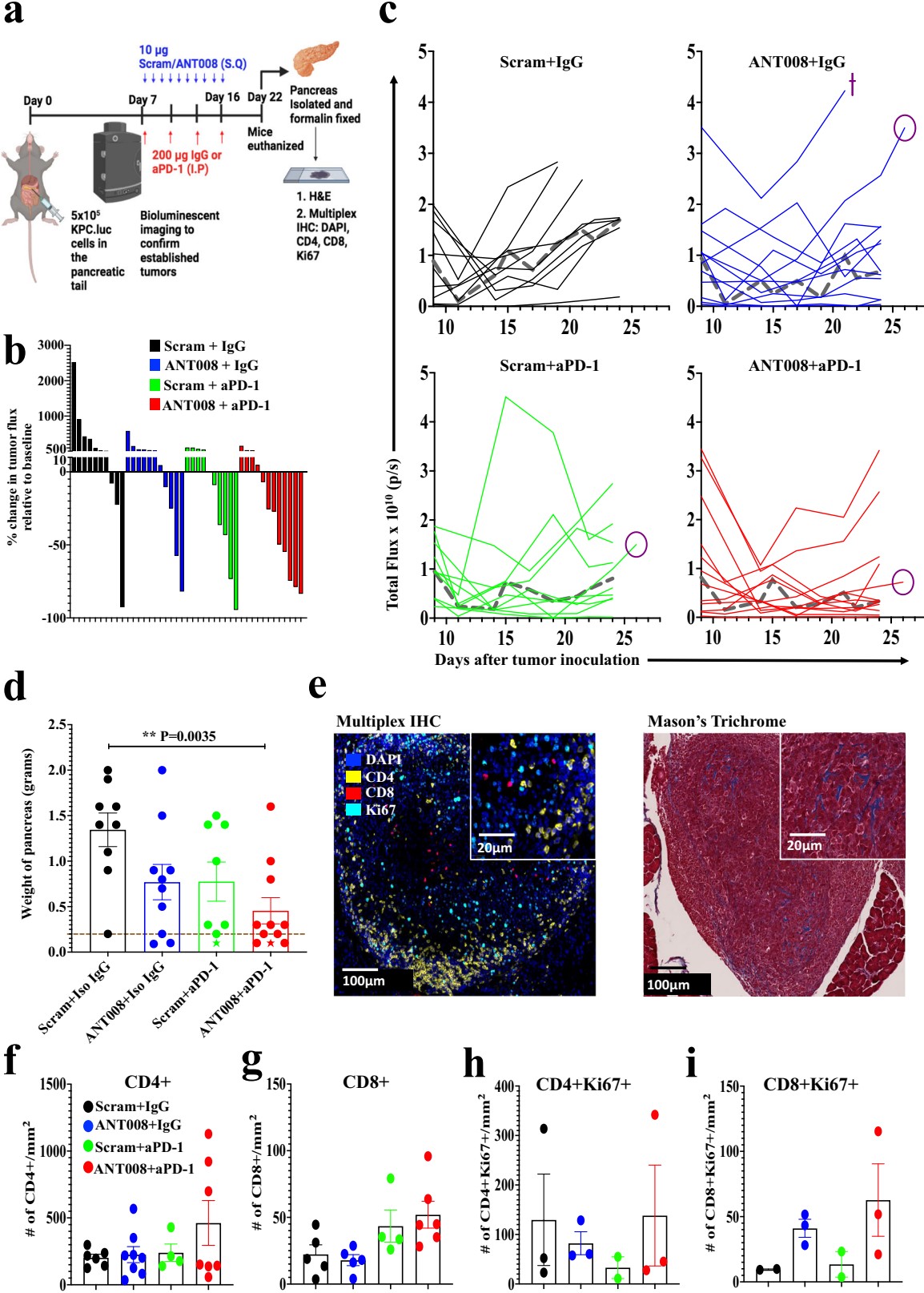

(Manassass, VA) and SM1 were from Dr. Antoni Ribas (UCLA, Los Angeles, CA). MT5 and BXPC3 cells were cultured in Roswell Park Memorial Institute (RPMI) medium supplemented with 5% and 10% Fetal Bovine Serum (FBS), respectively, in addition to 10mM L-glutamine and antibiotics. KPC-Luc, Panc02 and Panc1 cells were cultured in Dulbecco's Modified Eagle Medium (DMEM) supplemented

with 10% FBS, 10mM L-glutamine and antibiotics. Synthego (Redwood City, CA) provided CRISPR/Cas9 VPAC2 knockout pools of Panc02 cell line. Single clones of VPAC2 KO Panc02 cells were selected by limit-dilution and expanded in same media as wild type Panc02 cells. B16F10 cells were cultured in DMEM with L-glutamine and sodium pyruvate, supplemented with 10% FBS, 100ug/mL streptomycin, and 1500 mg/L

**Fig. 6 | Synergism between ANT008 and anti-PD-1 increases T cell infiltration and proliferation and decreases tumor burden in orthotopic KPC.Luc murine PDAC.** KPC.Luc cells were orthotopically implanted in the tail of the pancreas of C57BL/6 female mice and treated with ANT008 and/or anti-PD-1 with *n* = 9, 10, 8 and 11 in scrambled+IgG, ANT008 + IgG, scrambled+anti-PD-1 and ANT008 + anti-PD-1, respectively. **a** Schematic showing orthotopic implantation of KPC.Luc cells and treatment strategy with ANT008 and/or anti-PD-1 was created using BioRender. **b** Waterfall plot showing % change in tumor flux on day 22 relative to day 7 prior to start of treatment. **c** Total flux as measured by IVIS bioluminescent imaging in the different treatment groups. Isoflurane was used for anesthesia for bioluminescent imaging. Median flux represented as a dashed gray line (┅┅). Cross symbol (+) represents mice that were euthanized before day 25 due to ulceration of the tumor and circle symbol (○) represents the mice (one from each group) that were imaged

on day 26 via MRI imaging shown in supplementary figure S5. **d** Bar graph showing the weight of pancreas on day 25 when the mice were euthanized. 'Star' shaped (★) data points indicate tumor-free mice, and the dotted horizontal line represents the average weight of healthy pancreas from naïve mice. **e** Representative multiplex IHC images (right) showing pancreatic tumors stained for DAPI (blue), CD4 (yellow), CD8 (red), and Ki67 (cyan) and trichrome staining (left) with black arrows showing blue collagen stain in the tissue. The experiment was performed once with multiple groups stained at the same time. Bar plot showing number of (**f**) CD4+ or (**g**) CD8 + T cells/mm²; and (**h**) Ki67 + CD4 + or (**i**) Ki67 + CD8 + T cells/mm². *P* values shown in panel (**d**) were calculated using student ANOVA followed by Dunnett's post hoc test (comparing each treatment group with Scram + IgG). Error bars show mean ± SEM. **p < 0.01.

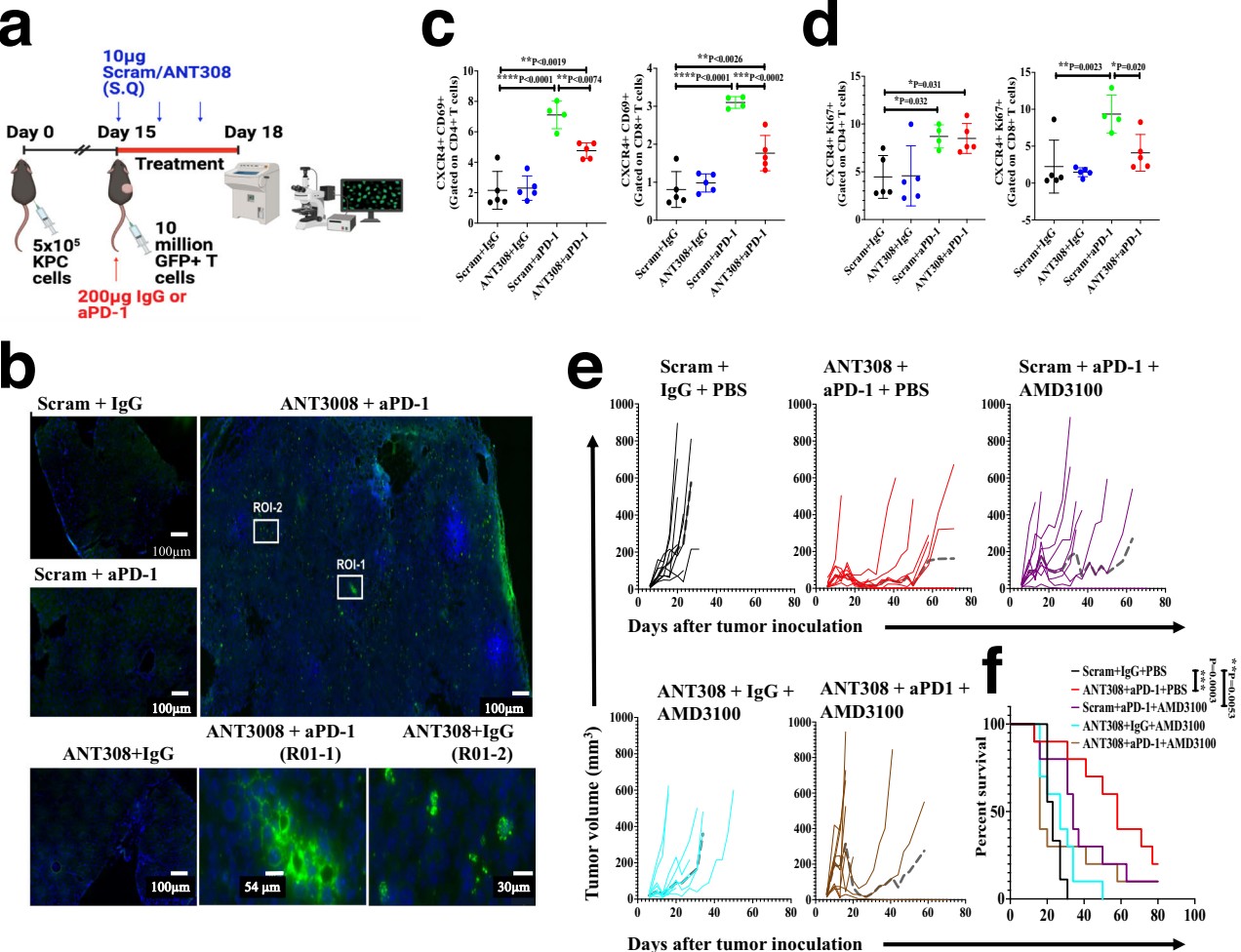

**Fig. 7 | Combination therapy with VIP-R antagonist and anti-PD-1 promotes intra-tumoral T cell infiltration and decreases CXCR4 expression on T cells in tumor-draining lymph nodes.** KPC.Luc tumors were implanted subcutaneously in C57BL/6 mice. On day 15 after tumor implantation, GFP⁺ T cells were adoptively transferred (via tail vein injections) and mice were treated with ANT308 ± anti-PD-1 for 3 days. **a** A schematic showing GFP + T cell transfer and treatment strategy in mice with subcutaneous KPC.Luc tumors were created using BioRender. **b** Representative Hoescht-stained tumor tissues (blue for nucleus) for each treatment group. Zoom-in of two regions of interest (RO1) labeled as ROI-1 and ROI-2 in the original image of tumors of mice treated with ANT308 + aPD-1 is also shown. The experiment was performed once with all four groups imaged on the same day.

Percentage of (**c**) CXCR4+ D69+ and (**d**) CXCR4 + Ki67+ cells in CD4+ (left) and CD8+ (right) subsets of T cells. **e** Tumor growth rate and (**f**) survival curves generated from mice with subcutaneous KPC.Luc tumors that were treated with scrambled peptide, IgG and PBS or ANT308 and aPD-1 or AMD3100 and aPD-1 or ANT308 and AMD3100 or a combination of ANT308, aPD-1, and AMD3100. Median tumor volume is represented as a dashed gray line. Statistical differences shown in panels (**c**, **d**) were determined via repeated measures ANOVA and Dunnett's post-test with *n* = 4–5 mice per group. Statistical differences in f were determined via a Log-rank test (*n* = 9–10 mice per group). Straight lines in panels (**c**, **d**) show mean ± standard deviation. *p < 0.05, **p < 0.01, ***p < 0.001, ****p < 0.0001.

sodium bicarbonate. The human pancreatic cancer associated stellate (PSC) cell line h-iPSC-PDAC-1 was generated and maintained as previously described[78]. Briefly, fresh pancreatic tissue was dissected into 0.5–1 mm³ pieces and plated in uncoated culture wells with Dulbecco's

Modified Eagle Medium (Gibco 11965-092) with 10% FBS (Gibco 26140-079) and antibiotics (Gibco 15240-062). These wells were incubated at 37 °C for 2–3 weeks until PSC were observed growing out of the tissue. Following 2 passages, pure PSC cultures were verified via morphology

and by α-SMA and GFAP staining. The h-iPSC-PDAC-1 cell line was processed as described above and was then immortalized via transfection with a lentivirus containing SV40 large T-antigen (GenTarget Inc.).

Pharmaceutical grade murine antibodies to PD-1 (Clone RMP1-14) or isotype control (Clone 2A3) were purchased from BioXcell (West Lebanon, NH). Pharmaceutical grade AMD3100 was purchased from Selleck Chemicals LLC (Houston, Texas). Scrambled peptide, ANT008 and ANT308 were purchased from RS synthesis (Louisville, KY) at a purity of >95%. 200 μM stock solutions were prepared in DEPC-Treated pyrogen free water from IBI Scientific (Dubuque, IA) and stored at −80 °C until use. Creative Biolabs (Shirley, NY) performed in-silico modeling to predict the binding affinity of ANT008 and ANT308 to VPAC1 and VPAC2 receptors.

## Patients and samples
Primary human PSC/CAF were isolated from resected pancreatic tumors in accordance with an IRB-approved protocol at the Winship Cancer Institute of Emory University on de-identified tumor tissues as previously described[79]. Briefly, freshly resected pancreatic tissue was dissected into 1 mm³ pieces, plated in a culture dish and incubated with DMEM + 10% FBS + antibiotics for 2–3 weeks until a coalesced cell monolayer was observed. Cell free supernatant was then collected to quantify levels of VIP via VIP specific enzyme immunoassay as described below. Blood samples from consented pancreatic cancer patients and healthy volunteers were collected in EDTA coated vacutainer tubes from Becton Dickinson and Company (Franklin Lakes, NJ). Plasma was isolated as previously described, where in the EDTA vacutainer tubes were centrifuges at 2000 x g for 15 min and stored at −80 °C until used for analysis of levels of VIP[80]. Patient demographics is described in Supplementary Table 1. Peripheral blood mononuclear cells (PBMCs) were isolated from patients with PDAC or healthy volunteers after informed consent (IRB 00087397 and IRB 00046063, respectively) by ficoll-hypaque density-gradient centrifugation as previously described and cryopreserved in CryoStor CS10 cell cryopreservation media (STEMCELL Technologies, Vancouver, Canada), until required for T cell isolation and ex-vivo expansion, as described below[81].

## Antibodies and flow cytometry
For western blot analysis of VPAC1, VPAC2, PD1 and CTLA4 expression, anti-VPAC1 (1:500), anti-VPAC2 (1:500) monoclonal antibodies from Sigma Aldrich (St. Louis, MO) and anti-PD1 (1:1000) and anti-CTLA-4 (1:500) monoclonal antibodies from Cell Signaling Technology (Danvers, MA) were used. For IF, anti-VIP monoclonal antibody at 1:50 from OriGene (Clone OT15B5) and anti-CK18 monoclonal antibody at 1:400 from Abcam (Clone EP1580Y) was used. Details of fluorochrome conjugated antibodies for flow cytometric analysis are provided in Supplementary Table 2. Fixable Aqua live/dead stain from Thermo Fisher Scientific (Waltham, MA) was used to detect and gate for live cells in all samples analyzed via flow cytometric analysis. To identify tumor specific T cells, APC conjugated MHC Tetramer H-2kb MuLV p15E from MBL International Corporation (Woburn, MA) was used. For intracellular cytokine expression staining, splenocytes or T cells were incubated with leukocyte activation cocktail (BD) for 5 h, then stained with antibodies listed in Supplementary Table 2. FACS files were acquired with a FACS Aria cytometer (Beckon Dickinson, San Jose, CA) or an Aurora cytometer (Cytek Biosciences, Inc, Fremont, CA) and analyzed using FlowJo software (Tree Star, Inc).

## Mice
Female or male C57BL/6, CD4KO (B6.129S2-Cd4$^{tm1Mak}$/J) and CD8KO (B6.129S2-Cd8a$^{tm1Mak}$/J) were obtained from the Jackson Laboratory (Bar Harbor, ME) at 6–8 weeks of age and housed in micro-isolator cages (stock #000664). Transgenic mice expressing enhanced green fluorescence protein (EGFP) from on a C57BL/6 background (strain designation: C57BL/6-Tg(Act-EGFP)C14-Y01-FM131 Osb) were a gift from Dr. Masaru Okabe (Osaka University, Osaka, Japan) and were bred and maintained at the Emory University Animal Care Facility (Atlanta, GA)[82] Experiments were performed when mice were 8-10 weeks old and animal care and maintenance was provided following *The Guide for Care and Use of Laboratory Animals* (National Research Council). Animals with tumor sizes exceeding the maximum (500 mm³) permitted under the animal use protocol (PROTO201700866) were humanely euthanized using $CO_2$. For CD4+ and/or CD8+ T cell depletion studies, antibody to deplete CD4+ T cells (Clone GK1.5) or CD8+ T cells (Clone 2.43) from BioXcell (West Lebanon, NH) were injected intraperitoneally at 200 μg per mouse on days −3, −1, +1, +3, +7 with respect to tumor implantation and twice a week until the completion of the experiment.

## Preparation of single cell suspensions
Harvested tumor tissues or tumor draining lymph nodes (TDLNs) from murine KPC-Luc or Panc02 bearing mice were cut into small pieces using a scalpel and treated triple enzyme digestion cocktail containing 10 mg/ml collagenase, 1 mg/ml hyaluronidase and 200 mg/ml DNase in HBSS at 37 °C for 20 min (TDLNs) or 1 h (tumors) and vortexed every 15 min. The tissue pieces were then mechanically dissociated, washed, centrifuged, and passed through a 70 μm nylon mesh filter to obtain a single cell suspension for staining and analysis via flow cytometry. For spleen samples, the single cell suspension obtained by mechanical dissociation, was passed through 70 μm nylon mesh filter and depleted of red blood cells using ammonium chloride lysis buffer and washed twice. Blood samples were collected in tubes with 0.1 ml diluted heparin (500 USP units/ml) followed by red blood cell depletion using ammonium chloride lysis buffer and washed twice.

## VIP specific enzyme immunoassay
$3 \times 10^5$ B16F10, KPC.Luc, MT5, Panc02, BXPC3 and Panc1 cells were cultured in six well plates with 3 ml of respective media. Cell free media was collected 24 h after culture and stored at −80 °C until tested for VIP levels. Peripheral blood collected from consented PDAC patients and healthy volunteers in EDTA tubes were centrifuged at 2000 g for 10 min to isolate plasma that was also stored in −80 °C until analysis. VIP levels in cell free supernatant and plasma were quantified via VIP specific enzyme immunoassay (EIA) kit following the manufacturer's protocol (RayBiotech, Peachtree Corners, Georgia). Absorbance was measured at 450 nm using synergy plate reader (BioTek, Winooski, Vermont), a standard curve generated and used to determine the concentration of VIP in the samples.

## Cell viability assay
To determine the effect of Ant-08 on the growth of PDAC cell lines in-vitro, MT5, KPC.Luc, Panc02, BXPC3, Capan-02 and cells were plated on a 96 well plate and treated with varying concentrations of Ant-08 (0–5 μM) in respective media, for 24–72 h. Cell viability was assessed using cell proliferation kit I from Roche (Basel, Switzerland), following the manufacturer's instructions. Briefly, at the end of the incubation time, 10ul of 0.5 mg/ml MTT labeling reagent was added and the plate was incubated in a humidified $CO_2$ incubator for 4 h. This was followed by incubation with solubilization buffer overnight and reading the absorbance at 570 nm. Percentage of cell viability with respect to control (0 μM ANT008) was plotted. Similar assay was performed to for wild type and VPAC2 KO Panc02 cells following treatment with Ant-08 or Ant-308 at 3 μm for 72 h.

## In vivo efficacy studies
For the subcutaneous model, $5 \times 10^5$ KPC.Luc cells were injected subcutaneously near the right flank of female or male C57BL/6 mice (stock

#000664). For the MT5 or Panc02 models, $5 \times 10^5$ were injected subcutaneously near the right flank of female C57BL/6 mice. For the orthotopic KPC-Luc model, mice were anesthetized, and the KPC-Luc cells were suspended in Matrigel and injected in the tail of the pancreas following laparotomy. 6–7 days after tumor implantation mice were randomized into four treatment groups and treated with VIP-R antagonist and/or anti-PD-1. While scram+IgG control mice received scrambled peptide and isotype IgG, the VIP-R antagonist, anti-PD-1 and VIP-R antagonist and anti-PD-1 groups received VIP-R antagonist and IgG; scrambled peptide and anti-PD-1; and VIP-R antagonist and anti-PD-1, respectively. The treatment regimen involved administering 10 µg of scrambled or VIP-R antagonist: ANT008 or ANT308, subcutaneously every day and 200 µg of IgG or anti-PD-1 intraperitoneally once every three days, for a total of 10 days. In experiments involving male mice, 20 µg of VIP-R antagonist was used due to the higher body weight when compared to female mice. In experiments were mice received AMD3100, 5 mg/kg of AMD3100 in PBS was administered subcutaneously every day for 10 days. In the KPC-Luc models, the tumor growth rate was plotted using the tumor flux measurements quantified using IVIS bioluminescent imaging. In the orthotopic KPC.Luc model, one mouse per group with biggest non-ulcerated tumor was sacrificed on day 28 and placed supine within a custom-built cradle and imaged with 9.4 Tesla MRI scanner. Rapid acquisition with relaxation enhancement (RARE) imaging sequence was used with a slice thickness of 0.4 mm and a total of 40 slices per mouse (RARE factor = 8, Average = 25 and Field of view = $30.7 \times 30.7$ mm$^2$). In experiments using mice with subcutaneous tumors, Vernier calipers were used to measure the tumor dimensions and the tumor volume was calculated using the formula: tumor volume = 1/2(length x width x height).

### Nanostring analysis of Tumor infiltrating T cells
Singlet cell suspensions of tumors were subjected to magnetic T cell isolation using EasySep™ Mouse CD90.2 Positive Selection kit II from STEMCELL technologies (Vancouver, Canada). Purity of T cells following isolation was 80–85%. Qiagen RNeasy Micro kit was used to extract RNA, and quantity and quality was assessed using the Nanodrop and Agilent 2100. RNA analyzed using the nCounter Metabolic Pathways Panel (Nanostring Technologies, Seattle, WA).

### TCR deep sequencing
Subcutaneously implanted KPC.Luc tumors treated with ANT008 and/or anti-PD-1 were harvested on day 21 after tumor implantation. Qiagen RNeasy Micro kit was used to extract RNA, and quantity and quality was assessed using the Nanodrop and Agilent 2100. Sequencing of TCR-β CDR3 V and J sequences performed by Adaptive Biotechnologies (Seattle, WA).

### Human T cell activation and expansion
One day before in vitro T cell expansion, cryopreserved PBMCs were thawed and rested by culturing at 37 °C in a 5% CO$_2$ humidified incubator overnight in complete RPMI media supplemented with 10% FBS, 100 U/mL penicillin and 100ug/mL streptomycin, MEM nonessential amino acids, 20mM N-2-hydroxyethylpiperazine-N-2-ethane sulfonic acid (HEPES), and 50uM 2-mercaptoethanol. T cells were isolated via negative magnetic isolation using EasySep Human T cell isolation kit from STEMCELL Technologies (Vancouver, Canada). 50,000–1 million T cells were seeded on each well of a 96 well plate that was coated with 10ug/ml of Ultra-LEAF Purified anti-human CD3 antibody (clone: UCHT1) from Biolegend (San Diego, CA) and cultured with 30U/ml recombinant human IL-2 (Peprotech, Inc., Cranbury, NJ) with or without 3 µM scrambled peptide, ANT008 or ANT308. Cells were counted using Trypan blue dye and phenotyped via flow cytometry on day 9. Every three or four days, the cells were split to 6 or 48 well or plates coated with anti-human CD3.

### Peptides
ANT008 has the peptide sequence of VIPhyb modified by replacing serine at amino acid position 25 with leucine. ANT308 is a further modification of ANT008 sequence in which the aspartic acid and asparagine residues at positions 8 and 9 are replaced with serine and aspartic acid, respectively. (ANT008: −60.17 kcal/mol free binding energy for VPAC1 and −51.07 for VPAC2, ANT308: −71.56 for VPAC1 and −56.27 for VPAC2 as per in silico analysis).

### Adoptive transfer of GFP + T cells
One million KPC-Luc cells were subcutaneously implanted into C57BL/6 mice. On day 15 after tumor implantation, spleoocytes from EGFP transgenic mice were harvested and processed as above and T cells were magnetically isolated using Pan T cell isolation kit II from Miltenyi Biotech (Auburn, CA). 10 million GFP + T cells were then injected intravenously to the KPC-Luc tumor bearing C57BL/6 mice. Mice were randomized into 4 treatment groups: scram+IgG, ANT308 + IgG, scram+aPD-1, ANT308 + aPD-1 and treated subcutaneously with 10ug of scrambled peptide/ANT308 on day 1, 2 and 3 and/or 200ug IgG or anti-PD-1 intraperitoneally on day 1 after T cell transfer. Mice were sacrificed on day 18 after tumor implantation and harvested tumors were flash frozen in OCT compound (Sakura Finetek, Torrance, CA) for embedding and cryosectioning. Tissue slides were then stained with 2ug/ml Hoescht 33342 (Abcam, Cambridge, MA) and imaged on a BZ-X810 epifluorescence microscope (Keyence Corp, Itasca, IL) using DAPI (359 nm/461 nm) and GFP (488 nm/510 nm) filter sets (Chroma Technology Corp, Bellows Falls, VT). Plan Fluor 40 × 1.3 NA oil immersion objective from Nikon Inc. (Melville, NY) were used with 100% of the light from excitation source reaching the sample, 1/1.2 s camera exposure and image aspect ratio of 1920 × 1440. Multiple images were acquired spanning the entire tissue section, which were then stitched using Keyence' image stitcher function and merged into a composite image using Fiji image analysis software.

### Western blots
Cells were washed twice with ice-cold 1X PBS and lysed with ice-cold RIPA (R0278, Sigma) containing 1X protease inhibitor cocktail (P8340, Millipore Sigma) and phosphatase inhibitors (P2850, Millipore Sigma). Lysates were quantified by Bradford assay (BioRad), normalized for concentration, denatured with 1XSDS sample buffer. 30µg of protein per sample was resolved by SDS-PAGE, blotted on PVDF membrane, and probed with primary antibodies described in Supplementary Table 2. The images were acquired using Azure 600 gel imager or Syngene G Box at autoexposure or exposure time between 1 and 2 min.

### Immunofluorescence
Paraffin-embedded PDAC tissues and adjacent normal tissues were deparaffinized, hydrated and antigen retrieved by boiling with 1XTrilogy for 15 min (Cell Marque-Trilogy Buffer) and subsequent washing with distilled water. Permeabilization was performed using 0.3% Triton-X-100, followed by blocking step with eBioscience™ low protein blocking buffer for 1 h at room temperature. Mouse anti-VIP (OriGene Technologies, Inc. Rockville, MD) diluted at 1:50 and rabbit anti-cytokeratin-19 (Abcam, Cambridge, MA) diluted at 1:400 was applied and incubated overnight at 4 °C. Secondary antibodies for anti-mouse IgG (H + L) conjugated with Alexa Fluor 647 and anti-rabbit IgG (H + L) conjugated with TRITC was applied and incubated for 1 h at room temperature. Tissue slides were then stained with 2ug/ml Hoescht 33342 (Abcam, Cambridge, MA) and imaged on a BZ-X810 epifluorescence microscope (Keyence Corp, Itasca, IL) using DAPI (359 nm/461 nm) and Alexa Fluor 647 (594 nm/633 nm) and TRITC (579 nm/599 nm) filter sets (Chroma Technology Corp, Bellows Falls, VT). Plan Fluor 40 × 1.3 NA oil immersion objective from Nikon Inc. (Melville, NY) were used with 100% of the light from excitation source

reaching the sample, 1/1.7 second camera exposure and image aspect ratio of 1920 × 1440. Multiple images were acquired spanning the entire tissue section, which were then stitched using Keyence' image stitcher function and merged into a composite image using Fiji image analysis software.

## Histology

All tissues for histology were fixed at 4 °C for 2–3 days in 4% paraformaldehyde in PBS, embedded in paraffin and cut into 5-μm thick sections. Slides were deparaffinized with EZ-Prep (# 05279771001, Ventana, Tucson, AZ) and antigen was then retrieved for 64 min with CC1 reagent (#950-500, Ventana, Tucson, AZ). Mouse anti-VPAC1 and anti-VPAC2 from Sigma Aldrich (St. Louis, MO) diluted at 1:500 or rabbit anti-cytokeratin-19 (Abcam, Cambridge, MA) diluted at 1:500 were applied and incubated for 40 min. DISCOVERY OmniMap anti-mouse or anti-rabbit HRP was applied and incubated for 12 min. The detection was completed in combination with DISCOVERY Chromo-Map DAB kit as per manufacturer's recommendations. Pancreas harvested from mice with orthotopically implanted KPC-Luc tumors or colon, liver and inflated lungs harvested from naïve C57BL/6 mice receiving ANT008 or ANT308 were formalin-fixed and paraffin embedded before being stained with H&E (Leica 560 MX, Wetzlar, Germany) or Masson's Trichrome (Polyscientific Inc., Bay Shore, NY). All slides were dehydrated, cover-slipped and scanned on Hamamatsu Nanozoomer 2.0 HT at 40x and reviewed by a pathologist.

## Multiplex Immunohistochemistry

Multiplex IHC staining was performed on the Roche Ventana DISCOVERY automated Immunostainer from Ventana Medical Systems (Tucson, AZ). The Ventana DISCOVERY uses a sequential staining procedure with a denaturation step between each staining sequence. Slides were deparaffinized with EZ-Prep (# 05279771001, Ventana) and then were antigen retrieved for 64 min with CC1 reagent (#950-500, Ventana). Cell Conditioning 2 buffer (CC2, #950-123, Ventana) was used for deactivation of the bound primary antibody and secondary anti-horse radish peroxide between each staining sequence. Four pre-diluted primary antibodies were sequentially applied in the following order using the indicated chromogenic detection: rabbit anti-Ki67 (#ab833 from Abcam, Cambridge, MA) with Opal 570, rabbit anti-Foxp3 (#NB100-39002 from Novus Biologicals, Littleton, CO) with Opal 480, rabbit monoclonal CD4 (#ab133616 from Abcam) with Opal 620 and rabbit anti-CD8 (#ab4055 from Abcam) with Opal 690 at dilutions of 1:300, 1:500, 1:500 and 1:250, respectively. Slides were cover slipped with VECTASHIELD Antifade mounting medium (Vector Laboratories) and stained slides were stored at 4 °C. Numbers of CD4+, CD8+, Ki67+ CD4+ and Ki67+ CD8+ T cells were quantified using QuPath, an open-source software for digital pathology image analysis.

## Statistics

For survival data, Kaplan–Meier method with log-rank tests was performed to determine statistical differences between treatment groups. For comparison of tumor volume and differences in immune cell subsets where four treatment groups were compared, one-way ANOVA followed by Dunnett's multiple comparison post-hoc test was used. All comparisons were two-tailed. For data from ex-vivo expansion of T cells from healthy volunteers, repeated measures ANOVA was used followed by Dunnett's post-hoc test. For data from expansion of T cells from PDAC patients were T cell phenotype and characteristics between scrambled or ANT008 treated were compared, pair wise student $t$-test was utilized. In Supplementary Fig. 8d-g, R-squared values were generated from a linear regression model. $P$ values less than 0.05 were considered significant. All statistical analyses were conducted using GraphPad Prism software, version 8.2 (GraphPad Software, Inc., San Diego, CA, USA).

## Reporting summary

Further information on research design is available in the Nature Research Reporting Summary linked to this article.

## Data availability

The data underlying the figures are provided as a Source Data file. All the other data supporting the findings of this study are available within the article and in the supplementary figures. Nano string data are available under GEO accession number (GSE213778) and TCR sequencing data are available under GEO accession number (GSE213482). VIP mRNA data are available on The Cancer Genome Atlas (TCGA). Source data are provided with this paper.

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

## Acknowledgements

The authors thank healthy volunteers and patients for blood and/or tissue samples. The authors also thank the shared resources at Emory University, namely the Emory Integrated Genomics Core (EIGC), Emory Flow Cytometry Core (EFCC), Cancer Animal Models Shared Resource (CAMS), Cancer Tissue Pathology Core (CTP), Biostatistics Shared Resource (BSR) and Integrated Cellular Imaging Core (ICI), that provided services or instruments at subsidized cost to conduct some of the reported experiments. BioRender was used to make Figs. 6a and 7a. The authors thank Dr. Sarwish Rafiq for carefully reviewing the manuscript and providing valuable edits and Dr. Yuan Liu from Georgia State University for helpful discussions. The data regarding VIP or PDL-1 mRNA levels in human tumors discussed here are based upon data generated by The Cancer Genome Atlas Research Network: https://www.cancer.gov/tcga. This work was supported in part by Katz Foundation funding and Emory School of Medicine Dean's Imagine, Innovate and Impact (I³) venture award to EKW and NIH R01 CA207619 awarded to SNT. Part of the cost for TCR sequencing was covered by the Young Investigator Award from Adaptive Biotechnologies to SR. Some of funding for the immunohistochemistry staining of tissues was covered by Winship Cancer Institute Development Discovery and Therapeutic Program Pilot funding to S.R.

## Author contributions

S.R. conceptualized the study, designed, and performed experiments, analyzed data, wrote the manuscript, and secured funding. TP designed some of the in vitro and in vivo experiments, analyzed data and reviewed the manuscript. J.M.L., S.W., R.D. and J.Z. assisted S.R. and T.P. in performing some of the in vivo experiments and in the analysis of related data. H.Z. confirmed CRISPR-Cas9 knockout of VIPR2 via RT-PCT and provided input in experiment design of in-vivo experiments with the KO Panc02 cells versus the WT Panc02 cells. B.W. and M.Y.Z. implanted KPC.Luc cells orthotopically, provided primary CAFs and reviewed the manuscript. M.C. isolated tumor draining lymph nodes and reviewed the manuscript. YL performed statistical analyses. S.G. analyzed murine tissues for signs of toxicity and edited the manuscript. B.R. analyzed human PDAC tissues and reviewed the manuscript. A.S.M., S.C., H.K., B.E.R. and A.B.F. provided input in experimental design and critically reviewed the manuscript. S.N.T. helped design experiments, provided funding, and reviewed the manuscript. GBL helped conceptualize the study, provided technical expertise and inputs, and critically reviewed the manuscript. E.K.W. conceptualized the study, edited and revised the manuscript, responded to reviewer comments, and provided funding.

## Competing interests

Intellectual property related to the use of peptide antagonists to vasoactive intestinal polypeptides to treat cancer is the subject of US patent applications with S.R., T.P., J.M.L., A.S.M., S.N.T., and E.K.W. listed as inventors. These patents have been licensed to Cambium Oncology, LLC. E.K.W., S.N.T., and S.C. are co-founders and have equity in Cambium Oncology. A.S.M. is a former employee of Cambium Oncology. A.B.F. is a current employee of Cambium Oncology. A conflict-of-interest management plan has been reviewed and approved by Emory University. The remaining authors declare no competing interests.

## Additional information

Sruthi Ravindranathan [1,2] ✉, Tenzin Passang [1,2], Jian-Ming Li [1,2], Shuhua Wang[1,2], Rohan Dhamsania [1,2], Michael Brandon Ware[1,2], Mohammad Y. Zaidi[1,2], Jingru Zhu[1,2], Maria Cardenas [3], Yuan Liu [2,4], Sanjeev Gumber[5,6], Brian Robinson[5], Anish Sen-Majumdar[7], Hanwen Zhang [1], Shanmuganathan Chandrakasan [8], Haydn Kissick [2,3,9], Alan B. Frey[7], Susan N. Thomas [10,11], Bassel F. El-Rayes[1,2], Gregory B. Lesinski [1,2] & Edmund K. Waller [1,2] ✉

[1]Department of Hematology and Medical Oncology, Emory University School of Medicine, Atlanta, GA, USA. [2]Winship Cancer Institute, Emory University, Atlanta, GA, USA. [3]Department of Urology, Emory University School of Medicine, Atlanta, GA, USA. [4]Rollins School of Public Health, Emory University, Atlanta, GA, USA. [5]Department of Pathology and Laboratory Medicine, Emory University School of Medicine, Atlanta, GA, USA. [6]Yerkes National Primate Research Center, Emory University, Atlanta, GA, USA. [7]Cambium Oncology LLC, Atlanta, GA, USA. [8]Aflac Cancer and Blood Disorders Center, Emory University, Atlanta, GA, USA. [9]Emory Vaccine Centre, Emory University, Atlanta, GA, USA. [10]Woodruff School of Mechanical Engineering, Georgia Institute of Technology, Atlanta, GA, USA. [11]Parker H. Petit Institute of Bioengineering and Bioscience, Georgia Institute of Technology, Atlanta, GA, USA. ✉e-mail: srra@outlook.com; ewaller@emory.edu

