## [Peer Review File · Nature Communications]

Targeting vasoactive intestinal peptide-mediated signaling enhances response to immune checkpoint therapy in pancreatic ductal adenocarcinomaREVIEWER COMMENTS

Reviewer #1 (Remarks to the Author):

The manuscript explore that VIP-R antagonist peptides can be effective means to resolve in the resistant of immune checkpoint blockade for PDAC ttherapy. . This study is novel approach and the proposed mechanism for T cell mediated anti-tumor response by VIP-R antagonist is quite convincing.You should mention about the comments below.

1. What kind of the stimulation induce overexpression of VIP in several cancers? Where is VIP secretion from?

2. Are there no VIP-Rs(VPAC1 and VPAC2) in tumor cells or others of PDAC? Please show VIP-R in human PDAC using immunohistochemical staining like the Fig. 1B.Please mention the possible reasons in the introduction.

3. Results

Fig 1C and 1D

P5 line8 and 10,

You should show results of the mean \pm SD or SEM.

Fig. 1F; How many samples in this experiment? One sample?

4. Methods

What type of anesthesia was used for the experiment?

Minor points.

5. Figures

- Fig. 1A is too small. I don't understand the species of solid malignancies in horizontal axis. You should show the intelligible figure.

- Fig. 1B the photo of VIP stained PDAC tumor is unclear. Please show the sharp image like the right image (CK19).

- In Fig.3c, the expressions of bar and asterisk are different from the others.

- Fig.4C: Are there so statistical significance between Scarm +IgG and ANT008 +IgG in PD-1+ Tim3 expressions? Is this really correct? I think that the standard deviation of "ANT008 +IgG" are large.

- In Fig. 5B, this figure is not easy to read. Please correct the figure using the large characters.

- Fig.5: Please unify the size of the asterisks.

- Is the vertical axis of Fig. 6I correct?

Minor points

1. P6 line 18 Figure? \Rightarrow Fig. 1E

2. References

You should write references based on the journal instruction.

Ex) "Eigler, D. M. & Schweizer, E. K. Positioning single atoms with a scanning tunnelling microscope. Nature 344, 524-526 (1990)".

Please confirm your references again.

No. 12 P2357-64. 2357-2364(2009).

Reviewer #2 (Remarks to the Author):

The manuscript, "Targeting vasoactive intestinal peptide receptor signaling enhances T-cell mediated anti-tumor response to immune checkpoint therapy in pancreatic ductal adenocarcinoma", by Ravindranathan et al, shows that VIP is overexpressed in PDAC, and tests the potential value of inhibiting VIP receptor (VIP-R) signaling might enhance anti-tumor immunity

in murine PDAC models. They conclude that VIP-R signaling is a targetable immune checkpoint pathway in PDAC.

MAJOR STRENGTHS

- Well organized paper.
- Various PDAC models (different PDAC cell lines, depletion studies, tumor challenge, and T-cell infiltration studies) support their hypotheses.
- Additionally, they show relevance of findings in human PDAC.

OTHER STRENGTHS

- Investigation into the effects of VIP-R antagonism on T cell exhaustion is convincing, but not all of the data include the outcomes from both VIP-R antagonists (Fig. 2).
- Re-challenge data (Fig. 6G) are striking, but is this confirmed with ANT308 as well?
- T cell infiltration data are convincing.

MAJOR WEAKNESSES

- Descriptions and/or data supporting the validity/specificity of their antagonists are lacking.
- Although the authors demonstrate PDAC cells and tumors highly express VIP in culture/blood plasma, they need to validate that VIP and VIP-R expression is specific to the epithelial cells because the IHC data are not very convincing (Fig. 1B).
- The authors do not demonstrate inhibition of VIP signaling by the novel antagonists presented in this paper or prove higher affinity of these antagonists compared to VIPhyb. Importantly, the manuscript does not investigate the impact of knockdown or knockout of VIP-R in at least one of the tested PDAC model cell lines to further confirm the specificity of the effects.
 - They do not describe decreases in downstream signaling of VIP following treatment with ANT008 and/or ANT308 to prove antagonism.
 - It would also be useful and potentially important to see validation of their findings in a VPAC knockdown model to rule out off target effects (both in vitro & in vivo).
- Figures 5F and 3B (KPC) use the same model and the same treatments, but the survival curves appear to be different in each figure. In Figure 3B they interpret the data to show improved survival with the combination treatment. However, in Figure 5F the treatment with ANT008 + aPD-1 as well as the treatment with aPD-1 alone resulted in 60% tumor free mice, which does not support this hypothesis. These discrepancies, as well as any differences between these two experiments should be better explained.

OTHER CONSIDERATIONS

- In Figure 1A, the authors examine VIP mRNA expression from TCGA data in different cancer types, but is that specific for PDAC, or for all pancreatic tumors? TCGA PDAC databases may contain pancreatic neuroendocrine neoplasms, which can skew the results. Also, the authors should include survival data about VIP since that is available from TCGA.
- In Figure 1G, the h-iPSC-PDAC-1 CAF line secretes just as much VIP as some of the other murine PDAC cell lines. Further investigation into the effects of VIP-R antagonism on CAF subpopulations could be useful in their PDAC models.
- Although some of the survival data demonstrate statistical significance (e.g., Fig. 3), the effects seem modest. Hence, VIP may be a component of productive anti-tumor immune response, but not a major driver of benefit. This needs to be discussed more completely.
- Although there may be a significant percent change in tumor flux (Fig. 6B), the total flux data from tumors in all treatment groups does not seem to differ, suggesting that many of the tumors are not growing.

- The authors should add representative images for T-cell and fibrosis staining from all treatment groups (Figure 6E) since, as the authors note in their discussion, PDAC is known for its dense, fibrotic stroma.
- The correlations in Figure 6F-I are not convincing, however other data in this figure and throughout the paper highly support the author's hypotheses.
- The authors should address if metastasis is observed in any of the models since the authors state that "VIP... where it was noted to promote growth and metastasis of tumors"?
- BxPC3 is not tested in all their in vitro experiments (e.g. BxPC3 is missing from Figure S1D).
- The authors state that ANT308 has a higher affinity for VPAC but frequently switch back and forth between ANT008 and ANT308 without describing the data for both antagonists or explaining why they decided to use one as opposed to the other.
- Did the authors investigate other immune cell populations in their models?

Reviewer One:

1. What kind of stimulation induce overexpression of VIP in several cancers, where is VIP secretion from?

Our data indicate that VIP is synthesized constitutively by a number of cancers, particularly those of gastro-intestinal origin. Looking at relative levels of VIP mRNA from CA BioPortal, we noted that pancreatic cancer, colon cancer, and stomach cancer had the highest relative levels of VIP. While VIP is made physiologically by neurons in the enteric plexus as well as by epithelial cells, we show in a revised Figure 1 that the adenocarcinoma cells of pancreatic cancer express high levels of VIP. In addition, we find that the cancer associated fibroblasts also secrete VIP suggesting that VIP is a general feature of the immunosuppressive tumor microenvironment in pancreatic cancer adenocarcinoma. With respect to the type of stimulus that may lead to overexpression of VIP in pancreatic cancer cells, our preliminary results suggest that IFN- γ could be driver of VIP upregulation in pancreatic cancer cells with two cell lines (KPC and Panc1) showing modest increase in VIP secretion *in vitro* compared to control treated cells (**Reviewer Figure 1**, shown below). Further investigation on this pathway is underway.

Reviewer Figure 1. Relative levels of VIP in culture supernatants collected from murine and human PDAC cell lines cultured for 24 hours with or without recombinant IFN- γ treatment at 10ng/ml.

2. Are there no VIP-Rs (VPAC1 and VPAC2) in tumor cells or others of PDAC? The reviewer asked for VIP-R staining in human PDAC using immunohistochemical staining analysis.

Consistent with findings in human and murine cell lines expressing VPAC1 and VPAC2 via western blot (Figure S1a-c), we have determined VPAC1 and VPAC2 are both expressed in PDAC tissue. We present these new data in supplementary Figure S1d and 1e.

3. Results.

The reviewer asked for error bars in Figures 1C, 1D, as well as in the text.

We include these in the revised Figure 1. The number of samples in Figure 1F are listed in the figure legend.

4. Methods. What type of anesthesia was used for experiment?

Isoflurane was used for anesthesia during bioluminescent imaging. We have included this detail in figure legends; Fig 6c and Fig. S7a.

5. Figures. The reviewer notes that Figure 1a was too small.

We have enlarged this in the revised manuscript and increased the font size listing the different tumor histologies that show VIP mRNA from the CA BioPortal. Figure 1b has been revised with new data on immunofluorescent staining that show, with increased clarity, co-expression of VIP and CK19 in PDAC tumor samples. The error bars in asterisks in Fig. 3c have been made uniform. We have added the error bars and clarified the statistics in the legends in Figure 4b. In Figure 5b, the font has been enlarged for easier readability. The size of the asterisks has been made uniform in Figure 5. Figure 6 f-i has been revised to include absolute number of cells per mm² instead of correlation graphs.

Minor points.

Page6, line 18, Figure 1E has been cited.

References have been adjusted per journal format.

Reviewer Two:

The reviewer asks whether tumor-rechallenge of mice that had regressed tumors after treatment with ANT308 has been done.

We include these new data showing combined results of experiments using ANT008 or ANT308, with 50% survival among the group treated with the combination of VIP-R antagonists and anti-PD1 antibody compared with 25% survival at day 80 post-tumor challenge treated with single-agent anti-PD1 antibody. Both groups of tumor-free mice were then rechallenged with the same PDAC tumor cell line. Tumor-bearing mice in remission after initial treatment with the VIP-R antagonists ANT308 (n=3) or ANT008 (n=5) combined with anti-PD1 were completely resistant to rechallenge with the same tumor, while all mice that achieved an initial remission after treatment with anti-PD1 (n=6) succumbed to rechallenge with the tumor cells (Figure 5f and 5g).

Reviewer Table 1. List of VIP-R antagonists (ANTs) with their predicted binding affinities to human VPAC1-R and VPAC2-R in comparison to VIP and VIPhyb. Tabulated percent survival of leukemic mice treated with antagonists and median survival in days are shown. SCRAM1 denotes the scrambled-sequence peptide control.

Peptide name	Docking Score VPAC1-R (Kcal/mol)	Docking score VPAC2-R (Kcal/mol)	Percentage of mice alive at day 60	Median Survival Time (MST) (days)
VIP	-65.8	-52.61	0%	30, n=10, p=0.1001
VIPhyb	-60.62	-51.007	5%	34, n=20, p<0.0001
SCRAM	-42.84	-37.102	0%	28, n=30, p=NS
ANT008	-60.17	-53.978	16%	35, n=25, p<0.0001
ANT308	-71.56	-56.27	40%	34, n=15, p=0.0009

f

g

The reviewer notes a major weakness is lack of data supporting the specificity of the antagonists.

We developed the ANT008 and ANT308 VIP-R antagonists based upon their predicted higher binding affinity to VPAC1 and VPAC2. We present herein **Reviewer Table 1** showing predicted binding affinity to tumor VPAC1 and VPAC2 based upon *in silico* modeling of peptide binding to the respective VIP receptors performed by Creative Biolabs. Higher binding affinity for these novel VIP-R antagonists was correlated with improved tumor-free survival in B6 mice bearing the C1498 leukemia cell line (**Reviewer Figure 2**). These data are the subject of a forthcoming manuscript, but we include them here for the benefit of the reviewer.

Reviewer Figure 2. Kaplan-Meier curve of leukemia-bearing mice treated with VIPhyb, ANT008, ANT308, and PBS control. Mice were injected with 1 million luc-C1498 acute myeloid leukemia cells through tail vein. Prior to treatment, leukemia was confirmed by bioluminescence imaging. On day 6, the mice were treated subcutaneously with either VIPhyb, ANT008 or ANT308 (10µg/mouse) once daily for seven days. The survival the mice were checked daily, body weight twice weekly, bioluminescence image once weekly. Survival differences between groups were calculated with the Kaplan-Meier log-rank test in a pair-wise manner.

The reviewer asks for clarification that VIP and VIP-R expression are specific to the PDAC epithelial cells.

As noted above in the response to Reviewer 1, we've revised Figure 1b with immunofluorescent staining showing co-staining of VIP and CK19 of formalin fixed and paraffin embedded PDAC tissue sections. We have supplemented immunohistochemical staining with VPAC1 and VPAC2 in human PDAC tissue suggesting the uniform expression pattern enriched in PDAC epithelia cells (Supplementary Figure S1d&e). We believe that this will address the reviewer's concerns.

The reviewer asks about the higher affinity of the novel VIP-R receptors compared to VIPhyb.

As noted above, we have tested these VIP-R antagonist peptides as single agents in mouse models of leukemia. We show in **Reviewer Table 1** above that ANT008 and, more significantly, ANT308 have higher predicted binding affinity to the human VPAC1

and VPAC2 receptors and that this higher predicted binding affinity to VIP-R is associated with higher single agent activities in a mouse leukemia model. As noted above, these data will be presented in a forthcoming manuscript focused on the activity of VIP-R antagonists in AML.

The reviewer asks about the effect of knockdown of a VIP-R in one of the tested PDAC cell lines to confirm the specificity of the effect.

We contracted with Synthego to knockout VPAC2 from the Panc02 cell line using CRISPR/Cas9 technology. We isolated single clones from the cell product returned by the company and confirmed that they lack the VPAC2 chain by genomic RT-PCR of the VPAC2 mRNA as well as lacking protein expression by western blot. These new data are included in supplementary Figure S2b-h. We tested the growth characteristics of the VPAC2 knockout Panc02 cell line *in vitro* and showed that it is similar to the wild-type cell line in Figure S2e. In addition, we cultured the VPAC2 knockout and wild type cell lines in the presence of ANT308 and found no effects on growth inhibition *in vitro* Figure S2f. VPAC2 KO cells had a slight delay in growth *in vivo* as compared to wild-type cells (Figure S2g&h). These data support autocrine effects of VIP signaling through the VPAC2 receptor in the Panc02 cell line and have reflected this important point in the revised manuscript Results and Discussion.

The reviewer questions the differences in survival curves between Figures 5F and 3B using the KPC cell line.

We note that Figure 3 used the ANT308 VIP-R antagonist while Figure 5 used the less potent ANT008 VIP-receptor antagonist. We have repeated the treatment of primary KPC-luc tumors experiment with ANT308 and show the combined results of experiments using ANT008 and ANT308 in Figure 5. Figure 5f shows higher levels of survival among mice receiving either ANT008 or ANT308 (with each drug given in combination with anti-PD-1 antibody) versus treatment with either VIP-R antagonist alone, anti-PD1 alone, or mice that received control treatment. Around day 100 following initial inoculation with tumor, surviving mice from experiments utilizing either ANT008 or ANT308 in combination with anti-PD-1, or mice treated with anti-PD1 alone were rechallenged with the same KPC-luc tumor cell line. A control group of immunologically naïve mice received the same number of KPC-luc tumor cells. Figure 5g shows 100% survival among mice treated initially with ANT008 or ANT308 in

combination with anti-PD-1 versus 0% long-term survival among mice that were initially treated with single agent anti-PD1 antibody. We note in the revised Discussion that these results are consistent with the generation of long-term protective anti-cancer immunological memory following treatment with the combination of ANT-R antagonist peptides and anti-PD-1.

Other considerations: The reviewer asks an important question regarding the cell of origin for the TCGA PDAC datasets.

We note that the TCGA has been annotated to include only pancreatic adenocarcinoma, excluding the neuroendocrine neoplasms. Furthermore, in Figure 1, we show higher expression of VIP on human samples of PDAC compared to adjacent normal tissues and show high levels of secreted VIP in condition media of both mouse and human PDAC cell lines as well as higher levels of VIP in mice bearing transplantable PDAC cancer. We include survival figures generated from the TCGA stratifying PDAC patients by median levels of VIP mRNA as an additional **Reviewer Figure 3** below. We note that there is no prognostic significance of VIP expression, such that VIP mRNA levels are not significantly correlated with survival among patients with PDAC. We interpret the absence of prognostic significance of VIP mRNA levels for this malignancy to be similar-to the situation in lung adenocarcinoma in which levels of PD-L1 expression are not prognostic for survival of lung cancer patients but are predictive for their response to anti-PD-1 clinical antibody therapy. These data are the subject of a manuscript under review describing the predictive and prognostic values of VIP and PD-L1 across a variety of cancers.

Reviewer Figure 3. Overall survival of cancer patients with high and low levels of PD-L1 and VIP expression. (A) lung adenocarcinoma (VIP n = 203, PD-L1 n = 239) (B) pancreatic adenocarcinoma (VIP n = 90, PD-L1 n = 90). High and low expression were determined by median mRNA values based upon data from TCGA.

The reviewer asks an important question regarding the role of cancer associated fibroblasts in the immunosuppressive tumor microenvironment of murine PDAC.

The immunofluorescent staining does indicate a low level of VIP expression in the noncancerous in the tumor stroma that would support the role of CAFs in creating an immunosuppressive microenvironment in Figure 1b. We highlight these data in the

revised manuscript and include appropriate references for the role of CAFs in supporting the immunosuppressive quality of the tumor microenvironment.

The reviewer questions the magnitude of the survival benefit in mice treated with the VIP-R antagonist.

We agree with the reviewer that blocking VIP-R signaling is likely only one element needed to achieve durable remissions in this difficult to treat malignancy. We've noted this in the revised discussion and respectfully point out to the reviewer that single-agent VIP-R antagonists led to 40% tumor-free survival in mice with the transplantable MT5 PDAC cell line and complete remissions of established PDAC in 20% to 30% of the other murine models of PDAC when given in combination with anti-PD-1 antibody. While these results are promising, we agree that additional work needs to be done to optimize the effects of blocking signaling through the VIP-R.

The reviewer asks whether the total flux was different among treatment groups in Figure 6.

We have amended Figure 6c to include a dashed line showing median levels of total flux and that clearly shows lower levels of total flux from residual tumor among surviving mice in the group that received the combination of the VIP-R antagonists and anti-PD-1.

The reviewer requests collagen staining from all treatment groups from the orthotopic model of KPC.

We present additional histological sections with Masson's trichrome collagen staining from these groups as requested in Figure S8c. Notably, there are dense bands of collagen and fibrosis in tumors from all treatment groups using the orthotopic KPC model as noted by the reviewer.

The reviewer questions whether the correlation between tumor size and members of infiltrating T-cells in the orthotopic model are convincing.

We have presented the data as absolute numbers of cells per mm² in Figure 6f-i. We moved the correlation graphs into Figure S8d-g.

The reviewer questions whether metastases are observed in the orthotopically-implanted PDAC mouse model.

Our coauthor and collaborator Dr. Lesinski has sectioned the liver of mice with orthotopic implantation of the KPC-LUC cell line and not found evidence for liver metastases. Of note, this tumor grows quite rapidly, and mice typically need to be euthanized by 24 days post tumor implantation because of the large tumor burden. It's possible that micro-metastases might be present but are below the limit of detection by routine H&E staining at this timepoint.

The reviewer notes that BxPC-3 is missing from Figure S1.

We have added BxPC-3 datapoints in our revised Figure S2a.

The reviewer asks why different VIP-R antagonists were tested, namely ANT008 versus ANT308.

As noted above, we selected these antagonists based upon their predicted binding to the human VPAC1 and VPAC2 receptors using in silico modeling. ANT008 was one of the first novel peptide sequences synthesized. We subsequently developed ANT308 as a more potent VIP-R antagonist based upon further modification of the ANT008 sequence. We have updated most of the data in the manuscript to reflect the effects of the more potent VIP-R antagonist in the murine PDAC model systems. We note in the revised method section the origin of these VIP-R peptide antagonists and their respective binding affinities to human VPAC1 and VPAC2 as well as their single agent activity in mouse models of leukemia.

The reviewer asks whether other immune cells have been studied in these model systems.

We note in Supplementary Figure S6b and 6c that percentages of most immune cells in the spleen was not significantly affected by treatment of ANT008 or ANT308 in naïve mice. We did observe a trend for decreasing numbers of MDSCs (Supplementary Figure S6c). We are currently investigating how VIP-R antagonists affect other immune cells including MDSCs in murine tumor models.

REVIEWER COMMENTS

Reviewer #1 (Remarks to the Author):

Authors were corrected according to reviewer's suggestion.
The study derived from these are consistent and sound.
i look forward to seeing it in published article.

Reviewer #2 (Remarks to the Author):

The authors have satisfactorily addressed my comments, though I have three remaining issues.

1. VCAM is proposed to be an immune checkpoint. However, as a soluble factor, it is perhaps best described as a tumor-protective immunomodulator.

2. The authors have gone to great lengths to support the bold and exciting idea that VCAM can be modulated to promote the tumor infiltration of activated T cells in PDAC. They do not fully consider the roles of NK cells in the PDAC TME, and the impact of VCAM on their roles in promoting anti-PDAC immunity.

3. While the data are generally supportive, the magnitude of impact of the therapeutic effects is modest in many of the models. This, combined with disappointing results when CXCR4 modulation is added, suggests that impactful clinical translation will be challenging. The authors should describe why they think their proposed approaches will overcome these challenges.

REVIEWER COMMENTS

Reviewer #1 (Remarks to the Author):

Authors were corrected according to reviewer's suggestion. The study derived from these are consistent and sound. I look forward to seeing it in published article.

Authors' response: *Thank you.*

Reviewer #2 (Remarks to the Author):

The authors have satisfactorily addressed my comments, though I have three remaining issues.

1. VCAM is proposed to be an immune checkpoint. However, as a soluble factor, it is perhaps best described as a tumor-protective immunomodulator.

Authors' response: *We assume that the reviewer mistyped "VIP" as "VCAM" or that the computer word-processing program auto-corrected "VIP" into "VCAM". We thank the reviewer for this comment and agree that VIP: VP-receptor signaling is distinct from the cell-surface bound ligands and cognate receptors that have been described to-date as immune checkpoints.*

We have edited line 44-45 in the abstract stating "VIP-R signaling is thus a tumor-protective immunomodulator that is targetable in PDAC" and in line 70 stating "we identified overexpression of VIP, an immunosuppressive neuropeptide, as a novel target for modulation of anti-cancer immune responses therapy in PDAC" and removed the references to VIP signaling as an "immune check-point pathway". Line 80 was edited to "We hypothesized that paracrine production of VIP by tumor cells within the PDAC TME functions like an immune checkpoint pathway that limits the antitumor activity of VIP-receptor-expressing T cells".

2. The authors have gone to great lengths to support the bold and exciting idea that VCAM can be modulated to promote the tumor infiltration of activated T cells in PDAC. They do not fully consider the roles of NK cells in the PDAC TME, and the impact of VCAM on their roles in promoting anti-PDAC immunity.

Authors' response: *We appreciate that NK cells play a role in tumor microenvironment, and that salutious effects of VIP-R antagonists on anti-cancer immunity might be mediated through the actions of NK cells. We refer the reviewer to our 2013 publication in PlosOne (cited as reference 23) in which mCMV infection led to increase levels of Interferon-gamma expression in NK cells from VIP KO mice or NK cells from wild-type mice treated with VIP-receptor antagonist VIPhyb (data shown in **Reviewer Figure 1**), and a second 2016 publication in Cancer Research in which transplantation of donor bone marrow and splenocytes from beige mice partially abrogated the graft-versus-leukemia effects of treatment with the VIP-receptor antagonist VIPhyb (added to the revised manuscript as reference 52; relevant published data shown in **Reviewer Figure 2**). We note the contribution of NK cells to the anti-cancer immune responses in PDAC-bearing mice*

treated with VIP-receptor antagonists in line 404 “Of note, earlier studies demonstrating increased interferon-gamma production by NK cells following treatment of mCMV-infected mice with a VIP-R antagonist [23] suggests that NK cells may also contribute to the anti-cancer immune responses seen in tumor-bearing mice treated with a VIP-R antagonist. Reduced graft-versus-leukemia effects seen in leukemic mice transplanted with allogeneic marrow and splenocytes from *beige* donors further supports the contribution of NK cells to the anti-cancer immune effects observed following treatment with VIP-R antagonists [52]”. We appreciate the suggestion to test the effect of VIP-R antagonists on NK cells in solid tumor models and plan to test that in the future.

Reviewer Figure 1.

Li et al. Plos One 2013 | Volume 8 | Issue 5 | e63381

Reviewer Figure 2.

Li et al. 2016 Cancer Res. 76(23): 6802–15.

3. While the data are generally supportive, the magnitude of impact of the therapeutic effects is modest in many of the models. This, combined with disappointing results when CXCR4

modulation is added, suggests that impactful clinical translation will be challenging. The authors should describe why they think their proposed approaches will overcome these challenges.

Authors' response: *We appreciate the reviewer's perspective on the effects of CXCR4 antagonism when given with the VIP-receptor antagonists. We have edited the manuscript to remove a sentence in the Results that was duplicative of a similar sentence in the Discussion (line 337) and edited the Discussion (line 388) to state "These data suggest that down-regulation of CXCR4, but not complete blockade, could be a superior therapeutic strategy when using a VIP-R antagonist with anti-PD1 to promote T cell trafficking and cytotoxicity within the TME. The apparent antagonism seen when treatment with a VIP-R antagonist was added to the combination of anti-PD1 and CXCR4 antagonists suggests that impactful clinical translation combining these three classes of drugs together would be challenging". In making these edits we noted that reference 37 was also listed as reference 48 and have removed the duplicate listing. Finally, to address the Reviewer's last comment we added the following sentence in the Discussion (line 463): "The ability of small peptide-based drugs to diffuse into the stromal-rich TME of PDAC and promote the *in situ* activation of cancer-specific T cells may be an advantage of targeting the VIP::VIP-receptor pathway in immuno-oncology". Further elaboration of the approach of using VIP-R antagonists is currently focused on improving the pharmacokinetics of these molecules.*

REVIEWERS' COMMENTS

Reviewer #2 (Remarks to the Author):

The authors have addressed my comments satisfactorily.